# Does management support drive sustained agile usage? a serial mediation model and cIPMA perspective

**Uthpala Wijesinghe**[1], **Vidara Mapitiyage**[1], **Chathurya Wickramarathne**[1], **Chamoda Wickramage**[1], **Krishantha Wisenthige**[2]*, **Chathuni Aluthwala**[1]

1 Department of Information Management, Sri Lanka Institute of Information Technology, Malabe, Sri Lanka,
2 Department of Business Management, Sri Lanka Institute of Information Technology, Malabe, Sri Lanka

* krishantha.w@sliit.lk

**Data Availability Statement:** All relevant data are within the manuscript and its Supporting Information files.

## Abstract

Agile software development is immensely popular in the industry, but most teams struggle to sustain its use. Human factors like management support, agile training, agile mindset, and team resilience are often neglected, hindering long-term success. However, research has not explored their underlying mechanisms in depth. Therefore, this study examines if management support impacts the sustained usage of agile methodologies within software development teams. It subsequently investigates the individual and serial mediating effects of agile training, the agile mindset, and team resilience on this relationship. Additionally, it compares the importance and performance of management support, agile training, the agile mindset, and team resilience in infusing agile practices. Finally, it determines these antecedents' necessity for the enduring success of agile application. Data collected from 391 agile software development professionals using a structured questionnaire. Partial-least-squares structural equation modelling, importance-performance map analysis and necessary condition analysis were used to investigate relationships. The findings underscore the pivotal role of management support in infusing agile practices. Agile training, mindset, and team resilience emerge as critical mediators, with a strong serial mediation effect. While management support is paramount, its practical implementation falls short within teams. All four antecedents are found to be necessary for optimal agile sustainment. Thus, this study significantly advances theoretical understanding by introducing a serial mediation model that elucidates their mechanisms in impacting agile infusion. It extends prior organisational-level findings to the team-level. The study's quantitative verification of qualitative findings strengthens their generalisability to a broader spectrum of teams. It pioneers in expounding the constructs' relative importance, performance and necessity, to offer actionable insights for agile practitioners. Finally, it provides methodological guidance to apply importance performance map analysis and necessary condition analysis in agile software development research.

**Funding:** The author(s) received no specific funding for this work.

**Competing interests:** The authors have declared that no competing interests exist.

## 1. Introduction

The contemporary landscape of software development (SD) is fraught with rising volatility, uncertainty, complexity, and ambiguity [1]. This is largely triggered by the accelerating pace of technological innovation, convoluted client demands, hyper-competitiveness and the escalating pressure to deliver greater value within stiffer deadlines [2–5]. Such a turbulent environment has necessitated that SD teams undergo a fundamental reorientation of SD practices. Thus, this has impelled a shift towards agile SD (ASD) [6–8]. The adoption of such methodologies supports teams' flexibility, adaptability and continuous improvement, bringing about a heightened ability to sense and respond to change [9,10]. A growing body of research has consistently highlighted the substantial benefits of ASD, such as enhanced efficiency, less time to market, improved quality and greater customer satisfaction [11–14]. Hence, efficient agile usage ultimately reinforces competitive advantage. Against this backdrop, agile methodologies such as Scrum, Kanban, Lean and Extreme Programming have emerged as the de facto standard of SD teams [6,15].

Nonetheless, an ASD team's journey towards realising the benefits of agile extends beyond the mere adoption of agile practices. While agile methodologies offer a promising approach to navigating the volatile, uncertain, complex, and ambiguous landscape of software development, their long-term success hinges on their sustained or enduring usage [14,16–18]. In other words, to effectively ensure the complete realisation of the benefits of agile methodologies, they should not only be adopted but also sustained within ASD teams. Such sustainment comprises the subsequent evolvement, assimilation and infusion of agile practices to suit the teams' operating contexts. This, termed 'sustained agile usage' (SAU), requires a concerted effort beyond surface-level utilisation. Yet scholars such as Gregory et al. [19] and Wang et al. [20] point out that ASD teams are currently facing immense challenges in this regard. Addressing these challenges requires a profoundly nuanced understanding of the complex interaction of the factors that contribute to SAU at the team level. Such an understanding will enable optimal resource allocation and stimulate effective strategy development in the pursuit of SAU within ASD teams. These insights hold crucial implications for managers and practitioners.

The sociotechnical theory [21] posits that the optimal performance of a system, such as an ASD team, hinges on the interplay between social and technical elements. The social subsystem is comprised of human factors like team members, management, attitudes, behaviours and interactions. These actively shape the holistic system by determining how technical tools and processes are used. This perspective, reinforced by the Input-Mediator-Output-Input (IMOI) model of teams, emphasises that interactions between such social elements lead to the development of team emergent states. In turn, these emergent states trigger team outcomes like SAU. Hence, it is evident that the efficacy of agile methodologies within teams is critically contingent upon its human factors and their interactions [5,10,22–26]. Despite this, their neglect by both scholars and practitioners [10,27,28] exacerbates challenges in sustaining agile practices within teams.

Recognising their pivotal importance, scholars have endeavoured to investigate the sociological antecedents of SAU. Prior studies specifically highlight the roles of management support (MS), agile training (AT), and the agile mindset (AM) as critical success factors for organisational sustainment of agile [14,24,29]. Strong MS is crucial for providing necessary resources, fostering a supportive culture, and encouraging continuous improvement. AT was found vital to impart technical/business knowledge and facilitate effective team dynamics. An AM, characterised by a willingness to improve, collaborate, take initiative, and focus on customer needs, was also deemed essential. Moreover, research emphasises the importance of team resilience (TR) in SD team success [30]. TR is a team's ability to withstand, adapt to and recover from adversity, with minimum impairment to its functioning [31,32]. SAU

necessitates continually evolving team practices, while maintaining optimal performance amidst unforeseen challenges. In such circumstances, TR emerges as a determinant between team success and failure [33–36]. However, studies show that MS can directly impact both AT and AM [37]. Effective AT, in turn, equips team members with the skills and knowledge to adopt an agile mindset and implement agile practices. Furthermore, MS, AT and AM can contribute to TR, by helping equip teams with the necessary skills and mindset [28]. Thus, the current study contends that MS is a key initial step in sustaining agile practices, and that MS, AT, AM, and TR are interconnected factors that exert individual and serial impacts on SAU.

Given the importance of agile infusion for combatting the challenges of the ASD landscape, it is crucial to probe how the key factors of MS, AT, AM and TR impact SAU from different logical perspectives. Practical applicability being paramount, researchers can harness combinations of advanced analytical techniques to gain actionable insights. A commonly used and widely potent technique for modelling complex relationships is partial least squares structural equation modelling [38,39]. In addition, ASD researchers and teams would largely benefit from the use of importance-performance map analysis (IPMA), that complements PLS-SEM by adding practical relevance to findings. IPMA visually contrasts the identified factors' (MS, AT, AM and TR) importance in achieving SAU, with their current performance levels within ASD teams. By doing so, it helps pinpoint the most critical factors that require prioritisation [40], allowing effective resource allocation. However, a construct, albeit exhibiting low impact based on PLS-SEM and IPMA results, may still be a necessary condition for achieving optimal SAU [41]. A necessary condition analysis (NCA) [42–44] adds a novel perspective to the current study's results, by assessing if a certain level of MS, AT, AM or TR is necessary for SAU to truly manifest [41]. Thus, this novel approach combining PLS-SEM, IPMA and NCA (styled cIPMA) [41] significantly enhances the rigor of the current study. Such nuanced and profound understanding of SAU bolsters its practical implications for ASD teams.

However, despite the pressing need for novel research on SAU on these fronts, the existing body of research is significantly plagued by numerous limitations. First, no known study has incorporated the added perspectives of PLS-SEM, IPMA and NCA (cIPMA) to the exploration of SAU, significantly curtailing the practical utility of existing knowledge. Next, the majority of studies focus on the adoption stage of ASD, neglecting the equally, if not more, important area of post-adoptive usage [19,20,25]. Furthermore, human factors accounting for a large portion of the challenges faced by teams remain under-explored within the already sparse domain of SAU. While MS, AT, and AM were qualitatively introduced as critical success factors of SAU, quantitative research is needed to verify these findings. Furthermore, as extant studies address the impact of these factors on SAU at the organisational level, failing to probe the factors at team level constitutes a critical oversight. Moreover, despite research conceding that TR is crucial in the context of ASD, very limited investigation of its impact on agile infusion exists. Additionally, the mechanisms through which MS, AT, AM, and TR influence SAU remain poorly understood. This is because limited research has explored the role of AT, AM, and TR as potential individual and serial mediators in the relationship between MS and SAU. Thus, these limitations form a critical knowledge gap that will severely impede the development of effective interventions to sustain agile practices.

The failure to develop effective interventions towards sustaining agile within the already challenging landscape of ASD would prove largely detrimental for teams. The incomplete application and benefit realisation of agile methodologies resulting from such knowledge scarcity could potentially lower team agility, decrease productivity and innovation, ultimately eroding competitive advantage. Therefore, to bridge these critical knowledge gaps and ensure the continued success of agile methodologies, our research aims to address the following five research questions.

RQ1 –Does management support have a significant positive impact on sustained agile usage?

RQ2 –Do agile training, agile mindset, and team resilience individually mediate the relationship between management support and sustained agile usage, and how important is each in the sequence?

RQ3 –Do agile training, agile mindset, and team resilience serially mediate the relationship between management support and sustained agile usage?

RQ4 –How does management support, agile training, agile mindset, and team resilience, perform compared to their importance in fostering sustained agile usage?

RQ5 –Are management support, agile training, agile mindset and team resilience necessary but not sufficient conditions for sustained agile usage?

This research offers a groundbreaking contribution to the broader field of ASD in three key ways.

First, it offers a significant methodological advancement of ASD research by pioneering the utilisation of cIPMA in the field. The combined application of PLS-SEM, IPMA and NCA allows for a more rigorous and comprehensive analysis of the complex relationships. This novel approach sets a new standard for ASD research and practice. By doing so, it provides a referential guide for future studies looking to examine similar phenomena within the arena.

Second, the study provides significant actionable insights for agile practitioners (both team members and managers) by incorporating the findings of PLS-SEM, IPMA and NCA (cIPMA). The incorporation of IPMA and NCA empowers teams to identify and address performance gaps, allocate resources effectively, mitigate risks and streamline decision making. The theoretical framework developed in the study can be used to design and implement targeted interventions to improve teams' SAU.

Third, the study provides a significant theoretical advancement within the domain of SAU. It pioneers in providing quantitative evidence for the necessity and impact of four critical success factors of SAU at the team level. The study incorporates three individual mediation effects and a serial mediating mechanism into a single unified framework. Thus, it provides a broad and complex comprehension of how the interplay of the necessary factors MS, AT, AM and TR ultimately lead to SAU. Furthermore, it makes a valuable addition to the literature on post-adoptive agile usage which is an extremely under-researched yet important area [14,20,25]. This is crucial considering that teams face significant challenges in this regard. It also makes huge strides in exploring human or sociological factors affecting SAU, an underexplored area within the wider domain of ASD.

The following subsections are organised as follows. Section two will provide a brief overview of the key scholarly works supporting the development of this study's hypotheses. Section three will outline how the study was carried out and section four will provide the results obtained. Section five will put the results into context using existing research, highlighting the theoretical and practical implications of the same. Finally, section six will provide the study's conclusion, and section seven will present the limitations and future research avenues enabled by the current study.

## 2. Literature review and hypothesis development

### 2.1. The concept of sustained agile usage

Agile methodologies are a response to a critical need for SD teams to be flexible, adaptable and continually competitive in the midst of numerous challenges. [6–10]. However, while the initial adoption of agile methodologies is a critical first step, this is insufficient to ensure the long

term realisation of agile benefits [14,18,20,25,45,46]. Bhattacharjee [16] argues that to make the best use of any innovation or methodology, it needs to be used enduringly and persistently while being adapted to suit the operating context. It is therefore implied that contextual optimisation of the practice is crucial. Without such sustained use, the application of agile methodologies within teams would remain primitive, highlighting the importance of the concept of SAU. However, teams often encounter obstacles in their efforts towards the same [18,20,29,45], constituting a significant business problem. Hence, making a clear distinction between adoption and sustained usage of agile practices is essential to comprehensively assess the implications of the same. Thus, the current study aims to make a valuable contribution to the understanding of ASD by investigating the post-adoption or sustained stage of agile methodologies.

The concept of SAU at the team level refers to 'the infusion of agile methodologies, practices and values in a team beyond adoption, alongside their continual improvement and evolvement' [14,18,19,27,45]. This is distinct from the initial adoption of ASD by a team in that its scope extends beyond simple use or disuse [14,16,20,29]. To build a sound theoretical basis for further research and exploration of the post-adoptive stage, Senapathi and Srinivasan [24] adopted the well-known sequential model of information system implementation [47,48]. This consisted of the stages of initiation, adoption, adaptation, acceptance, routinisation and infusion, out of which the latter three stages become the basis for post-adoptive or sustained agile usage. Acceptance refers to the stage where users commit to the use of an adopted information system, whereas routinisation refers to the stage where the use of the accepted innovation becomes normalised and encouraged. Infusion, the stage where SAU is primarily realised in, refers to the deep permeation and ingraining of the routinised innovation within the adopting entity (in this context, the ASD team). The infusion stage is further described by 05 infusion facets: extensive use, integrative use, emergent use, intensive use and deeply customised use. Extensive use refers to the widespread application of an agile practice to a broader range of work tasks. This suggests that the practice has become deeply integrated into the organisation's processes and is used to address a variety of tasks and processes. Integrative use refers to the establishment and strengthening of interlinks between work tasks, implying that the practice is not used in isolation but is seamlessly incorporated into the overall workflow. Emergent use is concerned with the application of an agile practice in ways that were not originally anticipated or intended. Next, intensive use denotes the application of an agile practice at a higher frequency or intensity than suggested by the standard guidelines. This may involve using the practice more frequently or applying it to more complex tasks. Finally, deeply customised use indicates the significant adaptation of an agile practice to reflect the unique needs and characteristics of the adopting unit or team. This involves tailoring the practice to fit the specific context and culture of the team, ensuring that it aligns with its overall goals and objectives [20,48]. Using this theoretical groundwork, SAU is defined at the organisational level as the amalgamation of two types of uses [14,24]. Vertical use, composed of the five facets of infusion, signifies the depth and intensity of agile practice adoption within a specific team or project. It encompasses factors such as the extent to which agile practices are integrated into daily operations, the level of customisation applied to these practices, and the degree to which team members have internalised agile principles and values. On the other hand, horizontal use refers to the breadth of agile practice adoption across an organisation. It involves the spread of agile practices to multiple teams, projects, regions, or departments. Horizontal usage indicates the extent to which agile methodologies have become a pervasive and institutionalised practice within the organisation [14]. However, this study's scope is limited to SAU at the team level and does not consider the broader implications of horizontal usage within the organisation.

It is evident that the arena of SAU presents a critical area of investigation for practitioners and academics alike, due to the need for the realisation of benefits offered by agile methods. Nonetheless, the scrutinisation of the extant literature on ASD reveals major limitations in their applicability. First, the underrepresentation of recent research focused on the sustained usage phase of agile methodologies is especially pronounced. While numerous studies have explored the initial adoption of agile practices [37,49–52], agile success [53] and agile team outcomes [54,55], there is a dearth of research into sustained usage of agile practices. In other words, research tends to address either the inputs or outputs of agile implementation, without investigating the journey of agile practice evolution. This oversight limits the understanding of true long-term agile optimisation. Furthermore, studies' lack of clear conceptual distinction between usage and sustained usage of agile implementation [17,25,56] further exacerbates the research gap. Many studies conflate these two stages, leading to vague conceptualisations and a failure to recognise the unique challenges and opportunities associated with each phase. Despite seminal work providing a theoretical foundation for understanding the post-adoption phase [14,24], subsequent research has largely ignored this conceptualisation [25,53]. This has resulted in a fragmented and inconsistent body of knowledge. Thus, the current study aims to advance the field by adhering to a rigorous conceptualisation of SAU, addressing the limitations of previous research.

The overreliance on case study and qualitative methodologies in the existing literature [25,29,57] also limits understanding of SAU, as well as practical applicability and generalisation. This methodological limitation, compounded by the lack of quantitative verification of the studies, hinders robust theoretical advancements and practical guidelines. Moreover, the perspective of the team, the primary work unit in ASD, is largely neglected in the already scant research domain. Multiple scholars examine the concept on the organisational level [14,18,25,29,45]. Thus, the resulting lack of unified and theoretically sound knowledge on effectively utilising agile would potentially lead to lost competitive advantage and reduced profitability of ASD teams. Such a background critically underscores the need for deep investigation into SAU. Thus, the current study will quantitatively investigate the post-adoptive stage of agile on the team level.

## 2.2. The human perspective of agile teams

The socio-technical theory of information system research [21,58] posits that any system, such as an agile team [59], comprises two interconnected subsystems: the social subsystem and the technical subsystem. The social subsystem encompasses the human elements, including team members, management, their interactions, behaviours, attitudes, team dynamics and cultural norms. The technical subsystem, on the other hand, consists of the tools, technologies, practices and processes employed within the system [60]. The two subsystems show a substantial reciprocal influence on each other, ultimately shaping the nature and direction of the holistic system [61]. However, from a sociotechnical perspective, the technical subsystem serves as a tool that empowers individuals and teams, rather than dictating their actions. This perspective emphasises humans as the active agents in shaping the technical component [62]. Therefore, successful change within a socio-technical system, such as the optimisation of agile, relies primarily on its social element [63]. Thus, this theory provides a strong theoretical foundation for assessing the human factors that influence agile adoption and sustainment.

Empirical evidence recognises the fundamental role of human factors in the success of agile practices [10,14,18,27,57,59,64,65]. The 'Agile Manifesto' [23] emphasises the importance of human factors over technical processes. As such, it is human-related challenges that pose significant barriers to the successful implementation and sustainment of a team's agile practices

[22,66]. Senapathi and Strode [29] elaborate that the path to agile sustainment is riddled with challenges related to the people involved and their lack of readiness to embrace agile values. They cite the lack of agile mindsets, leadership support, and unconducive team dynamics as factors. This aligns with Denning's [15] assertion that true agility requires a cultural shift, going beyond the mere adoption of practices to embrace agile values and behaviours. To be precise, teams are required to 'be agile, rather than do agile'. Therefore, insufficient scholarly and practitioner focus on such human factors significantly inhibit the optimisation of agile methodologies. However, despite the importance of human factors within the context, research providing strong evidence for the role of such factors in SAU is still in infancy. Hence, applying a socio-technical lens to the study of SAU, this research aims to investigate how specific human factors contribute to successful agile sustainment.

In their seminal work, Senapathi and Drury-Rogan [14] qualitatively introduced and refined a foundational framework for the critical success factors of organisational SAU. In this model, they underlined the pivotal roles of sociological influences. MS, by way of actively facilitating the implementation of agile through providing resources, training, creating a supportive culture and empowering teams, was found to be instrumental in fostering SAU. Furthermore, they elucidated the vital role of the AM. They asserted that its inherent characteristics such as the enthusiasm to learn, willingness to collaborate, proactive and independent initiative, and customer-centricity were central to ensuring agile assimilation in teams. Likewise, the importance of AT and mentoring of team members to this end was demonstrated to be significant. It was recognised that the imparting of technical and business knowledge pertaining to agile methodologies while facilitating agile team dynamics through such training is crucial for SAU. However, the study failed to address the perspective of the ASD team, instead investigating the relationships on the organisational level. Besides, subsequent research has failed to build upon this seminal study with recent empirical verifications. Furthermore, limited quantitative analysis has been carried out with regards to the impacts of the variables on SAU. Thus, the importance of the MS, AT and AM for team level SAU remain unresolved and requires further empirical investigation.

Moreover, researchers such as Cheng et al. [30] emphasise the importance of the ability of SD teams to effectively adapt to change, overcome challenges, and maintain their performance in order to achieve agile success. This coincides with the concept of TR which denotes the ability of teams to weather change and adversity with minimal functional detriment [28,54,64]. Since SAU requires ongoing change, it is reasonable to posit that TR is a prominent factor in the success of the same. However, the lack of prior research on TR in the context of SAU presents a significant opportunity to explore its impact on agile infusion. Accordingly, given their strong theoretical and empirical foundation in ASD literature, this study selected MS, AT, AM, and TR as key human factors to investigate their impact on SAU.

In accordance with the sociotechnical perspective [21,58], it can be argued that the human aspects within the system of an ASD team interact with each other. These interactions thereby impact the technical subsystem through critical dependencies. The Input-Mediator-Output-Input (IMOI) model of teams [67] provides a theoretical foundation to understand how these effects unfold. It illuminates that inputs, such as management activities, team training and team member attitudes, trigger the development of team processes and emergent states [26,54,56,57,68,69], such as team resilience. These processes and emergent states, in turn, gives rise to desired team outcomes, such as sustained agile usage in this context. It additionally elaborates that outcomes are further propelled by interactions between inputs, emergent states, and between inputs and emergent states. Thus, it is reasonable to posit that the factors of MS, AT, AM and TR impact SAU through dynamic interactions that have a serial impact on the same. However, a holistic investigation of these factors within a single study on agile infusion

is notably absent from the literature. Due to this, a cohesive framework that addresses the mechanisms through which these factors dynamically impact SAU is still lacking.

The existence of the discussed research gaps has significant detrimental effects on both academic and practical spheres. The lack of knowledge hinders the ability to draw definitive conclusions and develop evidence-based recommendations for teams in different contexts. Without a clear understanding of these mechanisms, teams may struggle to effectively implement and sustain agile practices. This underscores the need for further investigation into the factors that contribute to SAU and the development of practical strategies to support teams in their agile journey. By addressing the aforementioned limitations, this study aims to provide a more comprehensive and actionable framework towards understanding mechanisms through which human factors drive sustained agile success.

The following sections will further expound on the relevance and interplay of the factors of MS, AT, AM and TR in the context of SAU. It will then detail the formulation of the study's hypotheses.

**2.2.1. Management support and sustained agile usage.** Previous research has consistently demonstrated the significant impact of MS on various outcomes in the agile context such as project success [18,57,70] and agile transformations [71,72]. Specifically, several studies have demonstrated that MS plays a crucial role in fostering SAU. Senapathi and Srinivasan [24] and Senapathi and Drury-Rogan [14] cited MS as a key antecedent, as it provides the necessary resources, encouragement, and guidance for teams to effectively implement and sustain agile practices. This shows that MS has a significant impact on the development of an atmosphere of joint problem-solving and a team culture that fosters a sense of belonging, respect, and encouragement, by creating team psychological safety [14]. This, in turn, can facilitate the successful adoption and sustainability of agile methodologies. More recent studies such as that by Senapathi and Strode [29], have corroborated this finding clearly through contextual evidence. Adzgauskaite et al. [53], and Barros et al. [18] have found an impact of MS on agile success within a team. In line with this reasoning, Klünder et al. [37] posited that the top management provides a conducive culture for agile practices by assisting the ASD team to become autonomous, while motivating the team members to embrace agile values.

Given the significant role of the construct in providing resources, training, and a supportive culture, this study posits that MS is a critical foundational step in establishing and sustaining agile practices within teams. However, Senapathi and Srinivasan [17] were unable to quantitatively establish a statistically significant relationship between MS and agile usage from a sample of 114 agile practitioners. While this study provides a useful starting point, a larger sample size would be necessary to draw more definitive conclusions. Furthermore, while the study examined usage of practices, it did not address the nuances of SAU. As such, this study aims to address these conflicting findings in previous research. Therefore, based on the above arguments, we hypothesise the following for quantitative verification.

$H_1$: Management support has a significant positive impact on sustained agile usage.

**2.2.2. The individual mediating role of agile training in the impact of management support on sustained agile usage.** The refined model of critical success factors of SAU [14] identifies that agile coaching plays a monumental role in fostering SAU. An external or internal agile coach is tasked with providing stakeholders with the technical and business expertise towards understanding and implementing agile practices. This is echoed by Senapathi and Srinivasan [17], who found a significant positive relationship between agile coaching and agile usage. According to Klünder et al. [37], coaching aids the team formulate solutions to problems, transfer theoretical knowledge on agile methodologies, and assisting the team to stay

motivated. Therefore, the coach plays a key role in a team's correct and contextually relevant use of agile, thus enabling their long-term sustainability [28,73].

Interestingly, Klünder et al. [37] note the role of the management in facilitating access to agile coaching through providing resources and time. In this regard, the management's dedication to maintaining communication between the team and the coach is invaluable. The involvement of the managers is crucial in helping the coaches face challenges proactively and address resistance to the sustaining of agile within teams. Therefore, it is evident that management support has a significant impact on agile coaching.

While agile coaching is often cited as a valuable source of training, it is important to recognise that there may be other avenues through which ASD team members can acquire these skills. However, extant studies often overlook the comprehensive impact of AT on SAU, focusing primarily on how coaches help maintain practices. This study therefore aims to incorporate the various ways in which team members can access AT, beyond the traditional role of the agile coach. Thus, it expands the conceptualisation of agile coaching to accommodate aspects such as on-the-job training, formal education, and such [74], terming this agile training (AT).

Notably, Grass et al. [28] show that agile transformations warrant careful training of team members in the responsibilities of their roles and in the general context of agile methodologies. AT is clearly shown to help ASD teams use and progress the usage of agile practices [75,76]. Furthermore, it is clear that the management has a crucial role in providing access to AT resources [28]. However, the literature on agile implementation provides limited evidence regarding the mediating role of AT in the relationship between MS and SAU.

To address this research gap, the present study hypothesises that,

$H_2$: Agile training positively and significantly mediates the impact of management support on sustained agile usage.

**2.2.3. The individual mediating role of agile mindset in the impact of management support on sustained agile usage.** The AM refers to a proactive and customer-centric attitude towards work, encompassing a continuous pursuit of knowledge, open collaboration, self-directed decision-making, and a commitment to delivering continuous customer value [10,27]. Such a mindset corresponds to the embodying of agile values and is vital for thriving in dynamic or complex work environments. Senapathi and Drury-Grogan [14] identified that the AM, also characterised by adaptability, dedication, risk-taking, intrinsic motivation, and the capacity to learn and grow, is critical for SAU. They rationalised that when teams deeply embraced agile values and principles through such a mindset, they were more likely to maintain these practices over time, even in the face of challenges and resistance. Furthermore, the study by Senapathi and Strode [29] determined that people's resistance to change due to a lack of a proper mindset was detrimental to sustaining agility within an organisation. Thus, a well-developed AM is crucial for guiding effective adaptation and ensuring that agile practices align with the team's unique needs and goals [37]. Similarly, Ozkan et al. [27] argued that any agile initiative should be preceded by the inculcation of the appropriate AM. Manen and Vliet [77] suggested that cultivating a mindset of flexibility, collaboration, and a continuous learning orientation, is fundamental to the success of agile expansion and transformation initiatives in large organisations. Nevertheless, these studies show limited quantitative evidence.

However, Klünder et al. [37] remark on the essentiality of MS to enable a culture of experimentation and risk-taking within a SD team, which ultimately permits the flourishing of team members' AM. According to Denning [78], management decisions play a pivotal role in shaping an agile culture within an organisation, which can contribute to the development and sustenance of an AM. Indeed, Gouda and Tiwari [79] affirm that effective leadership plays a

crucial role in fostering an AM within teams. Leaders can promote creativity and innovation by encouraging open communication, facilitating employee development, and providing clear guidance and support. By establishing clear roles and responsibilities, communicating effectively, providing timely feedback, setting goals, and monitoring progress, leaders can create an environment that empowers employees to take initiative, experiment and contribute to the team's success. As such, Ozkan et al. [27] file such a leadership approach under critical success factors for an AM. However, while previous research has explored the importance of both MS and AM in fostering SAU, there is limited evidence specifically demonstrating the mediating role of AM in this relationship. This gap represents an opportunity for the present study to make a significant contribution to the field by providing empirical evidence to support this theoretical connection.

Furthermore, prior studies lack a strong theoretical conceptualisation of AM when examining its impact on SAU. To comprehensively assess the same in line with the latest theoretical developments, the current study adopted the conceptualisation introduced by Eilers et al. [10]. This framework views the same as a multidimensional construct composed of four key dimensions: 'attitude towards learning spirit', 'attitude towards collaborative exchange', 'attitude towards empowered self-guidance', and 'attitude towards customer co-creation'. The use of Eilers et al.[10]'s conceptual definition of AM strengthens the theoretical underpinnings of the study, allowing for a more rigorous and meaningful analysis.

Thus, the study hypothesises the following.

H₃: The agile mindset positively and significantly mediates the impact of management support on sustained agile usage.

**2.2.4. The individual mediating role of team resilience in the impact of management support on sustained agile usage.** TR denotes the ability of a team to recognise, withstand, adapt to, and overcome adversity or extensive change [31,32]. McEwen and Boyd [80] conceptualised TR as a higher-order emergent state comprising 07 dimensions: resourcefulness, robustness, perseverance, self-care, capability, connectedness, and alignment. Resilient teams demonstrate resourcefulness by mobilising their capabilities and prioritising goals effectively. They are robust, characterised by shared commitment, adaptability, and a proactive approach to challenges. Perseverance, or the ability to endure hardship and remain solution-focused, is another essential attribute. Self-care is demonstrated through effective stress management and maintaining a balanced work environment. Resilient teams are also capable, welcoming critical feedback and seeking support from others. Connectedness, characterised by encouragement, cooperation, and collaboration, is another key attribute. Finally, alignment, which involves recognising the efforts and successes of team members and maintaining a positive outlook, is an essential aspect of resilience.

Scholars provide empirical evidence supporting the notion that TR is a key factor in fostering agility within SD teams [64,81]. Likewise, according to Lengnick-Hall and Beck [82], a team's resilience capacity can be seen as a precondition to strategic agility. Indeed, a team that lacks resilience and the ability to adapt to challenges will struggle to maintain agile practices and achieve long-term success. A team's adaptive capability significantly impacts its agile innovative capacity [28,64], an important consideration in the context of sustaining agile methodologies. Therefore, TR is particularly crucial in the context of agile practices, which require teams to continuously adapt to change and meet evolving demands. However, the impact of TR on the specific outcome of SAU has not been probed in prior research.

Remarkable scholarly attention has been awarded to the antecedents of TR, wherein research has recognised the contribution of MS towards its genesis. Varajão et al. [83] posit

that effective leadership and MS are essential for fostering TR and facilitating the successful implementation of agile practices. By establishing a strong leadership style, promoting a culture of openness and conflict resolution, prioritising continuous improvement, and empowering team members, the management helps create a supportive environment that enables teams to overcome adversity, adapt to change, and achieve long-term success. Grass et al. [28] discovered that supportive management behaviour can significantly influence the empowerment dynamics within teams, ultimately leading to greater adaptability and resilience. On a similar note, McCann et al. [84] demonstrated that the development of the adaptive capacity of a team necessitates strategic leadership and a strong commitment from the management, a finding corroborated by Hartwig et al. [85]. McCann et al. [84] also stressed the need for the top management to afford generous attention to enhancing the resilience of teams, placing importance on the design of appropriate interventions for the same. These findings, however, remain quantitatively unverified. Furthermore, the specific role of TR as a mediator in the relationship between MS and SAU remains under-explored. Therefore, taking the evidence discussed into account, the study hypothesises the following.

H$_4$: Team resilience positively and significantly mediates the impact of management support on sustained agile usage.

**2.2.5. The serial mediation role of agile training, agile mindset and team resilience in the impact of management support on sustained agile usage.** The work of Klünder et al. [37] alludes to a potential serial effect of MS on AM, through the acting of AT. Their study provides evidence that MS can indirectly impact the AM by providing the necessary resources, encouragement, and opportunities for team members to receive AT. AT, in turn, is shown to foster the development of an AM by equipping team members with the knowledge, skills, and attitudes necessary for successful SAU. Intriguingly, Grass et al. [28] suggested that enhancing team members' skills by delivering technical and contextual training leads to their empowerment as well as operation upon agile values, ultimately enhancing their adaptive capacity or resilience. This adaptive capacity is further driven by activities of the top management that foster such skills and empowerment within the team, and the embedding of agile values and principles into the team culture. Furthermore, as discussed previously, such an adaptive capacity is shown to be paramount in ensuring that teams tune their agile practices innovatively, in order to effectively overcome stressors. This bears substantial evidence for a potential serial effect of MS on SAU, through the acting of AT, AM, and TR in sequence. However, this serial effect has not been tested empirically, thus risking overlooking the nuanced and complex relationships between the variables. Therefore, the present study hypothesises the following.

H$_5$: Agile training, agile mindset, and team resilience positively and significantly serially mediate the impact of management support on sustained agile usage.

**2.2.6. Necessary conditions of sustained agile usage.** Antecedent factors of a certain outcome become necessary conditions if they require that a minimum threshold is reached for said outcome to manifest [42]. In other words, if any removal or reduction of the antecedent below that threshold would prevent the desired outcome from materialising, then that antecedent can be considered a necessary condition. Such conditions are expounded in literature as critical factors or preconditions [86]. The evidence presented thus far emphasises the indispensable characteristic of the constructs of MS, AT, AM and TR towards nurturing SAU. The work of Senapathi and Srinivasan [24] and Senapathi and Drury-Grogan [14] clearly establish these factors as critical success factors for maintaining agile practices, asserting that the

absence of each would inhibit the realisation of SAU. Furthermore, Vijayasarathy and Turk [7] found that the lack of MS constituted a substantial barrier to agile implementation and use. Gandomani et al. [87] additionally state that inefficient training was a major reason for the failure of agile transformation projects. Moreover, without the appropriate AM that provides a strong foundation for the team's behaviour [88], practices may be misapplied, leading to unintended consequences or a loss of their intended benefits [14]. On the same note, Gallivan et al. [48], as well as Dikert et al. [71] state that poorer levels of attributes such as innovativeness and affinity for uncertainty (significant characteristics of an AM) place constrictions on the post-adoptive success of agile practices. Team resilience is concurred to be a significant necessary condition as well, given that it is a crucial factor in determining the success or failure of an agile team's pursuits and agility [33–36,82]. However, to the best of the authors' knowledge, no known study has explicitly evaluated if MS, AT, AM and TR constituted necessary conditions for SAU/placed bottlenecks for SAU.

Furthermore, Senapathi and Srinivasan [17] recognised their study's limitation in allowing conclusions on causality based on just the impact of the identified factors on agile usage. Such a limitation hinders a comprehensive understanding of what factors are indispensable in the ASD context. They noted that better conclusions can be derived through the manipulation of the levels of the study variables to see how they affect agile usage. Hence, to address these gaps, the present study hypothesizes the following.

$H_6$: Management support is a necessary condition for sustained agile usage.

$H_7$: Agile training is a necessary condition for sustained agile usage.

$H_8$: Agile mindset is a necessary condition for sustained agile usage.

$H_9$: Team resilience is a necessary condition for sustained agile usage.

Fig 1 presents the proposed research framework and the hypotheses $H_1$ –$H_5$ of the current study.

## 3. Materials and methods

### 3.1. Sample and data collection

First, the minimum required sample size of the present cross-sectional study was determined using the inverse square root method proposed by Kock and Hadaya [89], and the guidelines recommended by Hair et al. [38], that calculated the sample size based on the minimum expected path co-efficient, the significance level and statistical power. A statistical power of 80% and a significance level of 5% was employed. Anticipated path coefficients of 0.11 to 0.2 were determined based on the study by Senapathi and Srinivasan [3], which explored a similar conceptual framework related to agile usage. Thus, a sample size of approximately 155 was calculated for the current study. A stratified multi-stage cluster sampling approach was employed to select participants from a comprehensive list of agile software development companies in Sri Lanka. This was compiled using publicly available information on software development companies listed on the official websites of EDB Sri Lanka [90] and SLASSCOM [91]. The list of companies was stratified based on employee size into small to medium-sized enterprises (10–200 employees) and large-scale enterprises (over 200 employees). The classification aligns with the national policy framework for small and medium enterprise development put forward by the Sri Lankan Ministry of Industries and Commerce [92]. This framework defines small to medium-sized enterprises in the service sector as those employing fewer than 200 people. Companies, serving as clusters, were then randomly selected from each stratum. Within each selected company, a random sample of agile teams was chosen based on assigned team

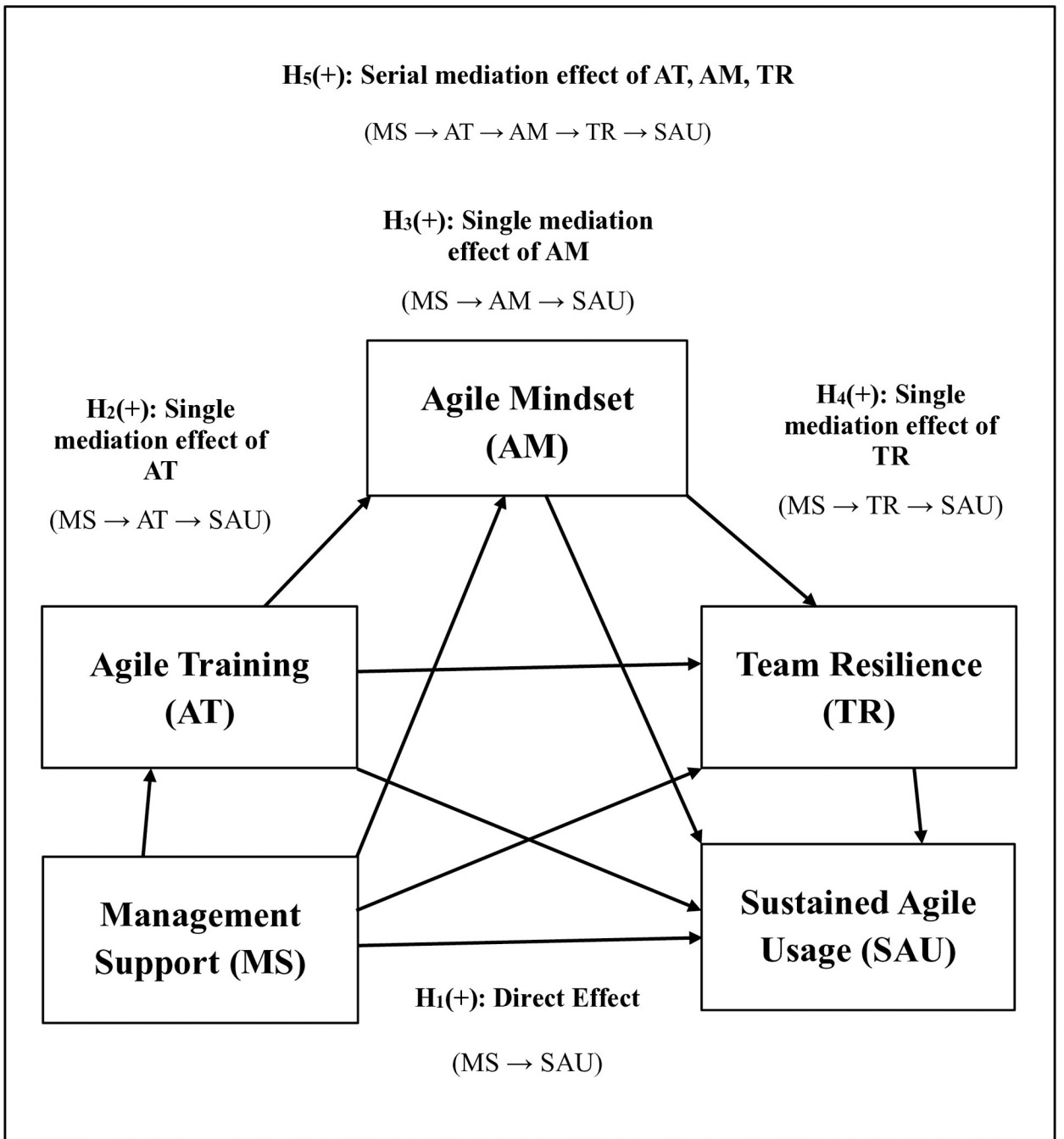

**Fig 1. Proposed research model for sufficient conditions.**

numbers. Finally, eligible ASD employees were randomly selected from each team using their employee numbers to form the final study sample. This multi-stage cluster sampling approach allowed for efficient and cost-effective data collection while ensuring adequate representation from different segments of the population.

Subsequently, a self-administered questionnaire was developed and pilot-tested on 45 selected agile software development professionals to ensure content validity. An agile expert

was additionally consulted during the development process. The pilot test and expert feedback revealed respondent concerns about survey length, ambiguous language, item redundancy and repetitiveness. Ambiguous items were rephrased to enhance clarity. Redundant and repetitive items were either removed or merged to streamline the survey [93]. Thus, the overall length of the survey was reduced from 84 to 51, retaining the most critical items to minimise respondent fatigue. After significant revisions, the improved survey was distributed online to agile software development team members with at least two years of overall experience in SD roles to guarantee maximum clarity and accuracy (Note that while their total experience might exceed two years, their experience in agile roles might be less, as their respective teams might revert to and from agile projects as required). A screening question (Yes/No) about whether ASD was used within respondents' current teams verified their eligibility. The data collection was conducted during the period from July 2024 –August 2024, during which 412 anonymous complete responses were received. While no missing data were reported, 21 responses were excluded due to a lack of variability within each response, indicating cases of straight lining. These responses were identified by a standard deviation of less than 0.25 within a single response. A final sample of 391 participants, far surpassing the minimum required sample size, was thus derived.

Male respondents accounted for 77.75% ($n_{male}$ = 304) of the sample, female respondents for 21.48% ($n_{female}$ = 84), while 0.73% ($n_{gender\ undisclosed}$ = 03) did not disclose their gender. This gender imbalance aligns with broader industry trends. The Global Gender Gap Report 2024 by the World Economic Forum [94] indicates that women represent only 37.64% of the overall ICT workforce in Sri Lanka and 28.2% globally in STEM fields.

Furthermore, a significant majority of 86.45% reported an experience of more than two years in ASD roles specifically ($n_{Agile\ experience\ >\ 2\ years}$ = 338). In terms of company size, 45.78% were from small to medium-sized enterprises, ($n_{from\ small\ to\ medium\ scale\ companies}$ = 179), while 54.22% were from large scale companies ($n_{from\ large\ scale\ companies}$ = 212). Team sizes varied, with 61.13% of respondents working in teams of fewer than ten members. In addition, 88.75% were aged 35 years and below ($n_{aged\ less\ than\ 35\ years}$ = 347). The responses showed that Scrum-based agile practices were dominant, with 96.93% of the sample using either Scrum or a Scrum-inclusive combination ($n_{using\ Scrum\ practices}$ = 379).

### 3.2. Ethical consideration

To ensure ethical consent, participants were provided with detailed information about the study's objectives and the questionnaire. All concerns were clarified prior to obtaining consent for participation. Explicit verbal consent was obtained from each respondent. Participation was entirely voluntary, and only those who explicitly consented were provided with access to the questionnaire. This consent process was witnessed by the authors. The study was conducted with the ethical approval of the Sri Lanka Institute of Information Technology (SLIIT) Business School Ethics Review Committee (SLIIT/ERC/SBS/2023/17).

### 3.3. Measurement instruments

In order to quantitatively measure the 05 variables of interest (MS, AT, AM, TR, and SAU), a total of 40 reflective measurement items were utilised. These items were either adapted from existing research with modifications to ensure contextual relevance or developed by the authors based on a thorough review of relevant literature. The development was further carried out in consultation with an agile expert as disclosed above, in order to identify which items and aspects to prioritise. This approach ensured that the measurement instruments were both theoretically grounded and practically applicable to the specific context of sustaining agile

software development within teams. Each item was to be rated on a consistent 05-point Likert scale ranging from "1 = Strongly Disagree" to "5 = Strongly Agree." 06 additional questions assessing the participants' demographic profile (age, gender, agile work experience, job role, company size and average team size) and 01 screening question determining their eligibility for the study was incorporated.

**3.3.1. Sustained agile usage.** The authors designed a section of 08 Likert-scale items intended to measure the extent to which the respondent's team has sustained its agile usage. The scale development was informed by the work of Senapathi and Srinivasan [17], and Senapathi and Drury-Grogan [14]. Two additional questions were added to assess the types and combination of agile methodologies used, and the types and combinations of agile metrics used within the team. Moreover, another question assessed if the team used agile methodologies in unconventional ways to support tasks other than software development. All three were not included in the final path modelling.

**3.3.2. Management support.** The scale for MS was developed based on the studies of Senapathi and Srinivasan [17], Senapathi and Drury-Grogan [14] and Klünder et al. [37]. 05 Likert-scale items were included, assessing the extent to which the management supported agile processes and the climate within teams.

**3.3.3. Agile training.** 05 Likert-scale items were again developed based on the studies of Senapathi and Srinivasan [17], Senapathi and Drury-Grogan [14] and Klünder et al. [37]. An additional question asking participants about the types of AT they have received was included as well.

**3.3.4. Agile mindset.** 10 items were adapted from the 20-item measurement tool for AM introduced by Eilers et al. [10]. The original scale was designed to assess four dimensions of AM: attitude towards learning spirit, attitude towards collaborative exchange, attitude towards empowered self-guidance, and attitude towards customer co-creation, and was reported by the authors to exhibit good to excellent reliability (all dimensions had exhibited Cronbach's alpha values of more than 0.71). Out of the adapted items, 02 items assessed the attitude towards the learning spirit while 03 measured the attitude towards collaborative exchange. Furthermore, 03 and 02 items measured the attitude towards empowered self-guidance and the attitude towards customer co-creation respectively. The items were carefully selected from the existing scale to maintain the construct validity and reliability of the original scale, while ensuring low redundancy and an appropriate context fit.

**3.3.5. Team resilience.** Finally, the 07 dimensions of team resilience (resourcefulness, robustness, perseverance self-care, capability, connectedness, and alignment) were measured using 12 items developed based on the 'Resilience At Work (R@W) Team Scale' [80]. 01 item was subsequently removed due to ambiguous cross loadings.

## 3.4. Common method bias

In line with the extensive recommendations provided by prior researchers [38,95,96], the authors strived to design, pilot-test and improve the scales such that common method bias was alleviated. Several procedural strategies were employed. First, only individuals with more than two years of experience in software development were included in the study, ensuring that participants were knowledgeable and engaged in the subject matter. Second, respondents were asked to reflect on their current team experiences, minimising the potential for retrospective recall bias. The study's relevance to ASD professionals was additionally emphasised to incite engagement and ensure that participants perceived their feedback as valuable. Next, to encourage honest and accurate responses, participants were made aware that their participation was voluntary and that their responses would be used solely for academic purposes. In addition,

existing scales were adapted and reworded to reflect the current study's needs. Incorporating feedback from the pilot-test, the total number of items was minimized from 84 to 51 (including demographic information) to alleviate respondent burden and survey abandonment. To address respondent concerns about survey length, item redundancy and repetitiveness, this study used single items to measure three sub-dimensions of the higher order construct of TR. This followed the guidelines of contemporary research [38,97–99], which approved the use of such items in cases of unidimensional narrow constructs. Examples were provided to clarify ambiguous items. The language used was kept simple.

A complete collinearity assessment is largely potent in evaluating the correlation among variables in a study [100]. A 'variance inflation factor' (VIF) exceeding 3.3 would provide a reliable indication of the presence of common method bias [101,102]. An assessment of the VIF values of the inner model in the PLS-SEM path model revealed that there were no critical levels of collinearity. The highest VIF value calculated was 2.749, confirming that the study was exempt from contamination by common method bias.

## 3.5. Statistical analysis

**3.5.1. Combined importance performance map analysis.** In order to ensure the fulfilment of the study's objectives, a multifaceted approach was employed. Complementing the results of standard partial least squares structural equation modelling (PLS-SEM) that assessed the structural model's relationships, the authors undertook an importance-performance map analysis (IPMA) to identify areas for improvement in sustaining agile usage. Subsequently, a necessary condition analysis (NCA) to test the necessity hypotheses was conducted. This combined approach, styled combined importance-performance map analysis (cIPMA) [41], is a recent addition to practical applications of advanced research methods, and allows deeper insight into the causal interplay of how MS, AT, AM and TR affect SAU. This permits greater practical applicability of findings [39]. The study followed the guidelines for the cIPMA as outlined by Hauff et al. [41] and Sarstedt et al. [103], further drawing on the recommendations of Richter et al. [104]. All analyses were carried out through the SmartPLS 4 software version 4.1.0.6.

*Partial least squares structural equation modelling (PLS-SEM)* is a muti-variate, causal-predictive approach to structural equation modelling (SEM) [105], that strives to maximise the explained variance of endogenous constructs. The employment of the technique in the current study is highly appropriate, owing to several advantages for complex research designs. Firstly, it allows scholars to comprehensively examine multiple constructs within complex model structures, making it particularly valuable for assessing latent and higher-order constructs [38,39,106–108]. This is a demand of the current study, which utilises the higher order constructs of AM and TR. Promising greater statistical power [108,109], PLS-SEM is well-suited for generalising findings to real-world contexts, seeing as it also imposes minimal distributional assumptions [38,39,110] on the data. This makes it a practical and popular choice for social and business research [39] where non-normal data is common [111]. Moreover, PLS-SEM's ability to bridge the gap between the dichotomy of prediction and explanation makes the technique relevant in the present study [108]. Thus, in line with contemporary recommendations of PLS-SEM [38,39], the authors adopted a two-step approach where they initially assessed the measurement models comprising of the lower and higher-order constructs [112,113]. The PLS-SEM algorithm was run with the path weighting scheme, to obtain standardised results. The authors then proceeded to assess the structural model comprising of the hypothesised relationships ($H_1$ –$H_5$). A significance level of 5% was used throughout (p-value < 0.05), and the bootstrapping procedure was used with 10,000 subsamples to test

structural relations. The bias corrected and accelerated bootstrap method (BCa) was used as the confidence interval method. The PLS Predict algorithm was additionally deployed with 10 folds and 10 iterations to assess the model's predictive validity as well as its out-of-sample predictive power [114].

To propel the formulation of further actionable insights for strategic initiatives centering on sustaining agile usage in teams, the authors subsequently undertook an *importance-performance map analysis (IPMA)* [40,115]. The simplicity of IPMA, as well as its pronounced applicability in identifying and prioritising immediate concerns within the landscape of SAU were prime considerations in adopting the analytical method. The analysis compared the exogenous constructs' importance in bringing about SAU (their total unstandardised effect or impact on the endogenous construct) against their actual performance in accomplishing the effect (an average of their unstandardised latent variable scores rescaled from 0–100) [38,40]. A single-unit increase in the performance of an exogenous variable increases the performance of SAU by the value of the concerned variable's total effect. The IPMA visually illustrates the above in the form of a two-dimensional map that shows construct's importance on the x-axis and their performance on the y-axis [41]. The map is demarcated into four quadrants along the collective average performance and average importance of all exogenous constructs. The upper-right quadrant holds the constructs that show high importance and high performance, whereas the upper-left quadrant displays the constructs that show low importance yet high performance. The constructs with low importance and low performance are shown in the lower-left quadrant. Here, the region of particular interest to the authors is the lower-right quadrant which presents the constructs that are pivotal to ensuring SAU (high importance), yet exhibit suboptimal performance, thereby highlighting critical areas that warrant immediate managerial attention [38]. Additionally, an extension of the same analysis to examine the individual indicators can help pinpoint the specific aspects of MS, AT, AM and TR that require targeted attention. The authors drew on these granular-level results to provide practical recommendations for practitioners. They utilised the importance-performance map analysis algorithm in SmartPLS 4, drawing on the results of the prior PLS-SEM analysis.

While PLS-SEM and IPMA are valuable tools, they operate predominantly on a sufficiency additive logic of causality. This focuses on assessing whether factors have a strong impact on an outcome and are 'sufficient but not necessary' to ensure the result. This perspective, although essential, might be incomplete. Solely relying on these techniques can therefore obscure a more nuanced understanding of the causal relationships involved. To address this concern, a *necessary condition analysis (NCA)* [42,43] was undertaken in order to test hypotheses $H_6$ –$H_9$. This analytical method adds value to the present study by incorporating a necessity logic of causality to determine if MS, AT, AM, and TR are 'necessary but not sufficient' to guarantee SAU [40]. In other words, the above constructs would be assessed to decipher if they impose bottlenecks on achieving a certain level of SAU (wherein increasing the values of other possible antecedent variables might not lead to improved performance unless the necessary condition's value is increased). It does this by establishing a ceiling line on top of the scatter plots of the exogenous constructs and the SAU construct, separating the area with observations from the area without observations. The greater the area without observations (the upper left corner), the larger the constraint or bottleneck placed by the exogenous variable on SAU, and hence the greater the effect size (d = the area without observations / the area with observations) [42,43]. In this study, three key parameters were considered: ceiling line accuracy (number of observations below the ceiling line as a percentage of the total number of observations), effect size (d) and effect size significance (p-value). The effect sizes were calculated using the unstandardised latent variable scores of MS, AT, AM, TR, and SAU calculated in PLS-SEM. The

**Table 1. Descriptive statistics and correlation analysis (Source: Authors' calculation).**

| Construct | Mean | Median | Standard deviation | Excess kurtosis | Skewness | SAU | MS | AT | AM | TR |
|---|---|---|---|---|---|---|---|---|---|---|
| SAU | 3.903 | 3.934 | 0.641 | 1.841 | -0.798 | 1.000 | | | | |
| MS | 3.809 | 4.000 | 0.869 | 0.419 | -0.792 | 0.642 | 1.000 | | | |
| AT | 3.952 | 4.000 | 0.709 | 0.728 | -0.594 | 0.527 | 0.605 | 1.000 | | |
| AM | 4.188 | 4.200 | 0.553 | 6.756 | -1.680 | 0.573 | 0.457 | 0.474 | 1.000 | |
| TR | 4.020 | 4.000 | 0.676 | 1.711 | -0.907 | 0.763 | 0.714 | 0.525 | 0.621 | 1.000 |

significance of the effect size was tested at 5% significance, using NCA's approximate permutation test [43] with 10,000 permutations in Smart PLS 4.

# 4. Results

## 4.1. Descriptive statistics

Table 1 provides a comprehensive overview of the descriptive statistics for the collected data. The analysis examines the central tendency, distribution, and correlations among the variables. The constructs of SAU, MS, AT, AM and TR show above-average means of 3.903, 3.809, 3.952, 4.188 and 4.020 respectively. This indicates that the teams, on average, exhibit strong levels of SAU, MS, AT, AM and TR. Thus, it implies that the teams potentially have adopted a strong foundation for agile principles and practices, a promising start to long term benefit realisation. Furthermore, the skewness and kurtosis parameters for each variable fall within acceptable thresholds of between -2.00 and +2.00, and between -7.00 and +7.00 respectively [116–118]. The boxplot in Fig 2 visualises the sample's descriptive statistics, providing a snapshot of the data's distribution. Next, the highest reported correlation between two factors is 0.763, highlighting the eligibility of the data to proceed with structural equation modelling.

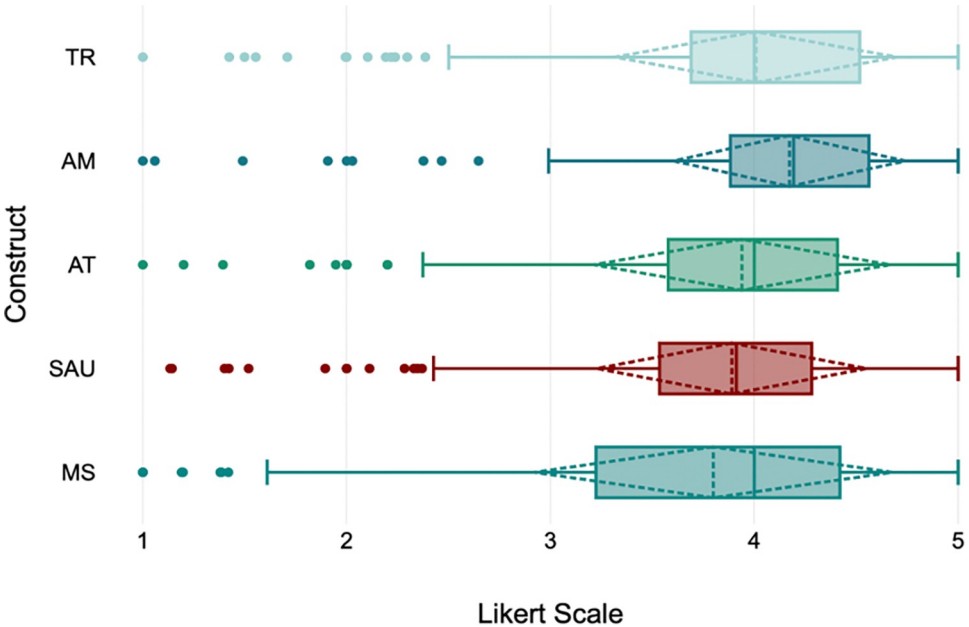

**Fig 2. Data distribution (Source: Created by authors using the Datatab application based on collected data).**

## 4.2. Results of partial least squares structural equation modelling

**4.2.1. Measurement model assessment.** The initial stage in PLS-SEM pertains to the measurement model assessment, which examines the relationship between the indicators and their constructs [38]. All constructs were modelled as reflective, in accordance with Coltman et al's [119] framework for determining the reflective or formative nature of a construct. The underlying latent variables of MS, AT, AM, TR and SAU exist independently of their measures, with variations in each construct causing variations in their item responses. Specifically, the items were observed to be manifestations of the underlying constructs, sharing a common theme and being interchangeable without altering their constructs' conceptual domains. Aligning with this, key studies that informed the conceptualisation and measurement of the variables in the current research have also employed reflective models [10,17,80]. Given the presence of the second-order variables of AM and TR, the study used the disjoint two stage approach of measurement model assessment [107,114,120]. This approach involved initially assessing the measurement model at the first-order level, followed by an evaluation at the second-order level. The construct validity was assessed at each level, using the indicator-level reliability, internal consistency reliability, convergent validity, and discriminant validity [38,39,118].

Table 2 presents the item-level, internal consistency reliability, and convergent validity results for first-order constructs. At the first-order level, the measurement model encompassing all separate sub-dimensions of AM and AT was validated. Item-level reliability was assessed using the outer loadings of each measurement item. Following the guidelines of Hair et al. [118], the conventional minimum threshold for an outer loading stands at 0.5, with a desirable value of 0.708. Items with loadings between 0.40 and 0.70 were contemplated for removal only if doing so resulted in a significant improvement in overall reliability and validity [38]. However, in this study, no such improvement was observed. Moreover, the reported lower-order outer loadings ranged from 0.634 to 0.942, exceeding the minimum threshold and demonstrating satisfactory item-level reliability (Table 2). Next, the internal consistency reliability of each lower-order construct was probed using Cronbach's alpha (CA) and composite reliability (CR) values [121], both of which should ideally fall between 0.70–0.95 [38,39]. However, in an instance when other construct validity measures yield satisfactory values, values between 0.60–0.70 are deemed acceptable [38,118]. It should also be noted that criticisms of Cronbach's alpha as being conservative and less accurate include its tendency to understate the reliability [39]. In comparison, CR tends to perform better with multidimensional constructs. With CA values ranging from 0.619 to 0.932, and CR values ranging from 0.814 to 0.948, the measures showed adequate internal consistency reliability at first-order level. Proceeding to the assessment of convergent validity, the average variance extracted (AVE) was utilised for the same. According to Hair et al. [39], a minimum AVE of 0.50 is considered acceptable, indicating that the construct explains at least half of the variance in its indicators. The current study's lower-order AVE values were calculated at a minimum of 0.569 and a maximum of 0.871, establishing its convergent validity at the lower level.

Next, the discriminant validity of the measures was verified using the Heterotrait-Monotrait ratio (HTMT ratio) [122], and cross-loadings. An item, Cap02, exhibited ambiguous cross loadings with another construct and was thus removed. As presented in Table 3, all items fell within the acceptable threshold of less than 0.90 [38,39], establishing discriminant validity at lower-order with a maximum value of 0.889.

Next, the latent variable scores generated by the Smart-PLS software were used in the creation and validation of the higher order measurement model. Table 4 shows the indicator loadings for each dimension of the two higher order constructs, all of which surpassed the threshold of 0.708. Both the CA and CR criteria achieved the minimum level of 0.70 for all

**Table 2. Measurement model assessment–lower order (Source: Authors' calculation).**

| Construct | Item code | Outer loadings | CA | CR | AVE |
|---|---|---|---|---|---|
| **Management support (MS)** | | | 0.932 | 0.948 | 0.786 |
| | MS01 | 0.885 | | | |
| | MS02 | 0.859 | | | |
| | MS03 | 0.915 | | | |
| | MS04 | 0.900 | | | |
| | MS05 | 0.872 | | | |
| **Agile training (AT)** | | | 0.918 | 0.939 | 0.755 |
| | AT01 | 0.784 | | | |
| | AT02 | 0.880 | | | |
| | AT03 | 0.915 | | | |
| | AT04 | 0.896 | | | |
| | AT05 | 0.863 | | | |
| **Agile mindset (AM)** | | | | | |
| **Attitude towards learning spirit (ATLS)** | | | 0.745 | 0.887 | 0.797 |
| | ATLS01 | 0.895 | | | |
| | ATLS02 | 0.890 | | | |
| **Attitude towards collaborative exchange (ATCE)** | | | 0.755 | 0.859 | 0.671 |
| | ATCE01 | 0.833 | | | |
| | ATCE02 | 0.851 | | | |
| | ATCE03 | 0.771 | | | |
| **Attitude towards empowered self-guidance (ATESG)** | | | 0.658 | 0.814 | 0.594 |
| | ATESG01 | 0.781 | | | |
| | ATESG02 | 0.695 | | | |
| | ATESG03 | 0.830 | | | |
| **Attitude towards customer co-creation (ATCC)** | | | 0.619 | 0.824 | 0.705 |
| | ATCC02 | 0.942 | | | |
| | ATCC01 | 0.723 | | | |
| **Team resilience (TR)** | | | | | |
| **Alignment (ALI)** | | | 0.852 | 0.931 | 0.871 |
| | Ali01 | 0.933 | | | |
| | Ali02 | 0.934 | | | |
| **Capability (CAP)** | | | | | |
| | Cap01 | 1.000 | - | - | - |
| **Connectedness (CON)** | | | | | |
| | Con01 | 1.000 | - | - | - |
| **Perseverance (PER)** | | | | | |
| | Per01 | 1.000 | - | - | - |
| **Resourcefulness (RES)** | | | 0.865 | 0.918 | 0.788 |
| | Res01 | 0.886 | | | |
| | Res02 | 0.905 | | | |
| | Res03 | 0.871 | | | |
| **Robustness (ROB)** | | | 0.761 | 0.893 | 0.807 |
| | Rob01 | 0.884 | | | |
| | Rob02 | 0.912 | | | |
| **Self-Care (SEL)** | | | | | |
| | Sel01 | 1.000 | - | - | - |
| **Sustained agile usage (SAU)** | | | 0.890 | 0.913 | 0.569 |

(*Continued*)

**Table 2.** (Continued)

| Construct | Item code | Outer loadings | CA | CR | AVE |
|---|---|---|---|---|---|
| | SAU01 | 0.764 | | | |
| | SAU02 | 0.852 | | | |
| | SAU03 | 0.827 | | | |
| | SAU04 | 0.763 | | | |
| | SAU05 | 0.799 | | | |
| | SAU06 | 0.709 | | | |
| | SAU07 | 0.634 | | | |
| | SAU08 | 0.660 | | | |

constructs at the higher order, establishing the internal consistency reliability of the measurement instrument. Furthermore, AVE values for all constructs exceeded 0.5, proving the higher-order model's convergent validity.

HTMT values below 0.9 of the higher order model showed the distinctive nature of the higher-order constructs in the study, establishing their discriminant validity. Therefore, based on the analyses conducted thus far, the measurement model is concluded to be satisfactory.

**4.2.2. Structural model assessment.** The second stage of PLS-SEM pertains to the assessment of the inner model or the structural model. First, the inner model's VIF values were examined to identify any potential collinearity issues that might inflate the results. Since all values were below the desired threshold of 3.3 [39], it was concluded that collinearity was not an issue in the current study.

Afterwards, the structural relationships ($H_1$ –$H_5$) were assessed at 5% significance, where a hypothesis was rejected if its p-value was greater than 0.05 (insignificant), or if its t value was below 1.96 [38]. All results of hypothesis testing are presented in Fig 3 and Table 5.

First, it can be observed that there is a significantly positive total effect of MS on SAU ($\beta$ = 0.654, p < 0.001, t = 18.626), showing that the former is an important antecedent of the latter. Investigating the links in detail, the direct positive effect of MS on SAU in the presence of AT, AM and TR ($\beta$ = 0.140, p = 0.007, t = 2.709), was statistically significant, substantiating $H_1$. Accordingly, the total positive indirect effect through the mediators had a path coefficient of 0.514 and was statistically significant at p < 0.001 (t = 11.918). It should be further noted that

**Table 3. Discriminant validity—HTMT ratio (Source: Authors' calculation).**

| | MS | AT | ATESG | ATCC | ATCE | ATLS | RES | ROB | PER | SEL | CAP | CON | ALI |
|---|---|---|---|---|---|---|---|---|---|---|---|---|---|
| **AT** | 0.666 | | | | | | | | | | | | |
| **ATESG** | 0.461 | 0.500 | | | | | | | | | | | |
| **ATCC** | 0.430 | 0.500 | 0.729 | | | | | | | | | | |
| **ATCE** | 0.504 | 0.527 | 0.763 | 0.736 | | | | | | | | | |
| **ATLS** | 0.495 | 0.505 | 0.810 | 0.669 | 0.881 | | | | | | | | |
| **RES** | 0.758 | 0.597 | 0.587 | 0.575 | 0.709 | 0.745 | | | | | | | |
| **ROB** | 0.657 | 0.588 | 0.561 | 0.601 | 0.615 | 0.584 | 0.889 | | | | | | |
| **PER** | 0.624 | 0.478 | 0.552 | 0.438 | 0.584 | 0.574 | 0.789 | 0.749 | | | | | |
| **SEL** | 0.534 | 0.377 | 0.431 | 0.319 | 0.439 | 0.398 | 0.568 | 0.607 | 0.525 | | | | |
| **CAP** | 0.632 | 0.424 | 0.438 | 0.401 | 0.494 | 0.521 | 0.738 | 0.675 | 0.679 | 0.584 | | | |
| **CON** | 0.603 | 0.451 | 0.389 | 0.476 | 0.490 | 0.504 | 0.730 | 0.770 | 0.616 | 0.579 | 0.680 | | |
| **ALI** | 0.738 | 0.565 | 0.493 | 0.556 | 0.648 | 0.620 | 0.886 | 0.811 | 0.747 | 0.579 | 0.740 | 0.837 | |
| **SAU** | 0.715 | 0.614 | 0.584 | 0.514 | 0.630 | 0.676 | 0.865 | 0.732 | 0.723 | 0.531 | 0.663 | 0.685 | 0.750 |

Table 4. Measurement model assessment–higher order (Source: Authors' calculation).

| Construct | Outer loadings | CA | CR | AVE |
|---|---|---|---|---|
| **Agile mindset (AM)** | | 0.832 | 0.888 | 0.665 |
| **Attitude towards learning spirit (ATLS)** | 0.850 | | | |
| **Attitude towards collaborative exchange (ATCE)** | 0.859 | | | |
| **Attitude towards empowered self-guidance (ATESG)** | 0.786 | | | |
| **Attitude towards customer co-creation (ATCC)** | 0.764 | | | |
| **Team resilience (TR)** | | 0.927 | 0.942 | 0.698 |
| **Alignment (ALI)** | 0.878 | | | |
| **Capability (CAP)** | 0.838 | | | |
| **Connectedness (CON)** | 0.854 | | | |
| **Perseverance (PER)** | 0.844 | | | |
| **Resourcefulness (RES)** | 0.886 | | | |
| **Robustness (ROB)** | 0.825 | | | |
| **Self-Care (SEL)** | 0.713 | | | |

the indirect effect accounted for 78.59% of the total effect, while the direct effect constituted 21.41%, demonstrating the vital roles of the mediators in the relationship. Probing further, all single mediating effects (indirect effects) of AT ($\beta = 0.069$, p = 0.028, t = 2.202), AM ($\beta = 0.038$, p = 0.021, t = 2.312) and TR ($\beta = 0.272$, p < 0.001, t = 7.390) were similarly proved positively significant, establishing proof for $H_2$, $H_3$ and $H_4$. Given that the direct effect of MS on SAU was significant and positive, all three mediating variables exhibited a complementary partial mediating effect in the aforementioned relationship. AT, AM and TR separately accounted for 13. 42%, 7.39% and 52.92% of the total indirect effect, respectively. Accounting for 7.98% of the total indirect effect, the serial mediation effect of AT, AM and SAU also played a noteworthy role in the same ($\beta = 0.041$, p < 0.001, t = 4.198), evidencing $H_5$. The findings thus verify

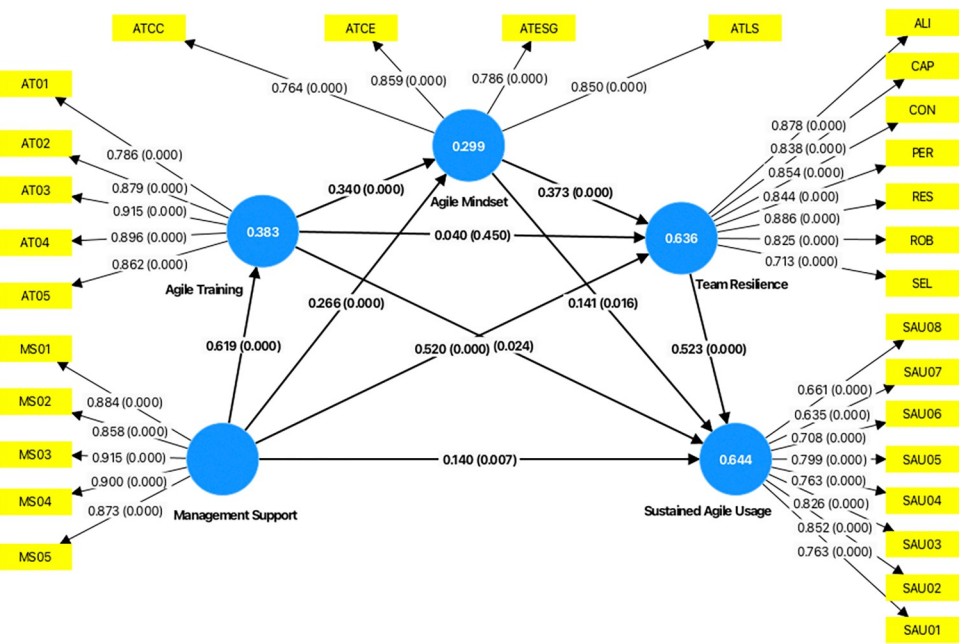

**Fig 3. Measurement and structural model assessment.**

**Table 5. Structural model results of H1 –H5 (Source: Authors' calculation).**

| Hypothesis | Structural path | Path coefficient (β) | Mean | Standard deviation | t-value | p-value | CI | | Decision |
|---|---|---|---|---|---|---|---|---|---|
| | | | | | | | Lower (2.5) | Upper (97.5) | |
| | **Total effect** | | | | | | | | |
| | MS → SAU (c) | 0.654 | 0.655 | 0.035 | 18.626 | < 0.001 | 0.578 | 0.716 | |
| | **Direct effect** | | | | | | | | |
| $H_1$ | MS → SAU (c') | 0.140 | 0.140 | 0.052 | 2.709 | 0.007 | 0.037 | 0.241 | Supported |
| | **Total indirect effect** | 0.514 | 0.514 | 0.043 | 11.918 | < 0.001 | 0.429 | 0.599 | |
| | **Single mediation effect** | | | | | | | | |
| $H_2$ | MS → AT → SAU | 0.069 | 0.068 | 0.031 | 2.202 | 0.028 | 0.013 | 0.134 | Supported |
| $H_3$ | MS → AM → SAU | 0.038 | 0.037 | 0.016 | 2.312 | 0.021 | 0.009 | 0.073 | Supported |
| $H_4$ | MS → TR → SAU | 0.272 | 0.272 | 0.037 | 7.390 | < 0.001 | 0.203 | 0.348 | Supported |
| | **Serial mediation effect** | | | | | | | | |
| $H_5$ | MS → AT → AM → TR → SAU | 0.041 | 0.041 | 0.010 | 4.198 | < 0.001 | 0.025 | 0.065 | Supported |

that MS furthers the SAU of SD teams, both directly, and indirectly through the single as well as serial mediation of AT, AM, and TR. Besides, the serial mediation effect of AT, AM, and TR on SAU further indicates a complex and interconnected relationship between these factors, making a compelling case for their nested importance within the context of ASD.

The co-efficient of determination ($R^2$) describes the amount of variance of an endogenous construct explained by its exogenous predictors, affording a measure of the in-sample predictive power (explanatory power) of a structural model [38,39,123]. Cohen [123] described $R^2$ values for endogenous variables as such: 0.26 as substantial, 0.13 as moderate, and 0.02 as weak. However, as explained by Chin [124], in the context of PLS-SEM, $R^2$ values of approximately 0.67, 0.33 and 0.19 can be interpreted as substantial, moderate, and weak respectively. In the present study, the variables MS, AT, AM and TR accounted for 64.4% of variance in SAU ($R^2$ = 0.644), proving a substantial explanatory power (Table 6). Also, AT, AM and TR boasted $R^2$ values of 0.383, 0.299, 0.636 respectively. The adjusted $R^2$ value of SAU was still substantial at 64%. Next, to assess the predictive relevance/accuracy of the structural model, the $Q^2$ metric was computed [125]. As evidenced by Table 6, the results demonstrated that all endogenous variables possessed significant predictive relevance, given that all $Q^2$ values exceeded zero [126]. $Q^2$ values also provide a scale for assessing the level of predictive relevance of the model, with 0–0.25 representing small relevance, 0.25–0.50 representing medium relevance, and above 0.50 representing high relevance [38,39]. As per the results obtained, AM exhibited small predictive relevance ($Q^2$ = 0.218). AT and SAU demonstrated medium relevance with $Q^2$ values of 0.377 and 0.423 respectively. TR displayed the highest predictive relevance ($Q^2$ = 0.516).

Next, to assess the out-of-sample predictive power of the model, the study compared the root mean square errors (RMSE) of prediction for the indicators of the key endogenous construct (SAU) in the PLS path model to that of a naïve linear model (LM) benchmark. They

**Table 6. Explanatory power ($R^2$) and predictive relevance ($Q^2$) (Source: Authors' calculation).**

| Construct | $R^2$ | Adjusted $R^2$ | $Q^2$ |
|---|---|---|---|
| **AT** | 0.383 | 0.381 | 0.377 |
| **AM** | 0.299 | 0.295 | 0.218 |
| **TR** | 0.636 | 0.633 | 0.516 |
| **SAU** | 0.644 | 0.640 | 0.423 |

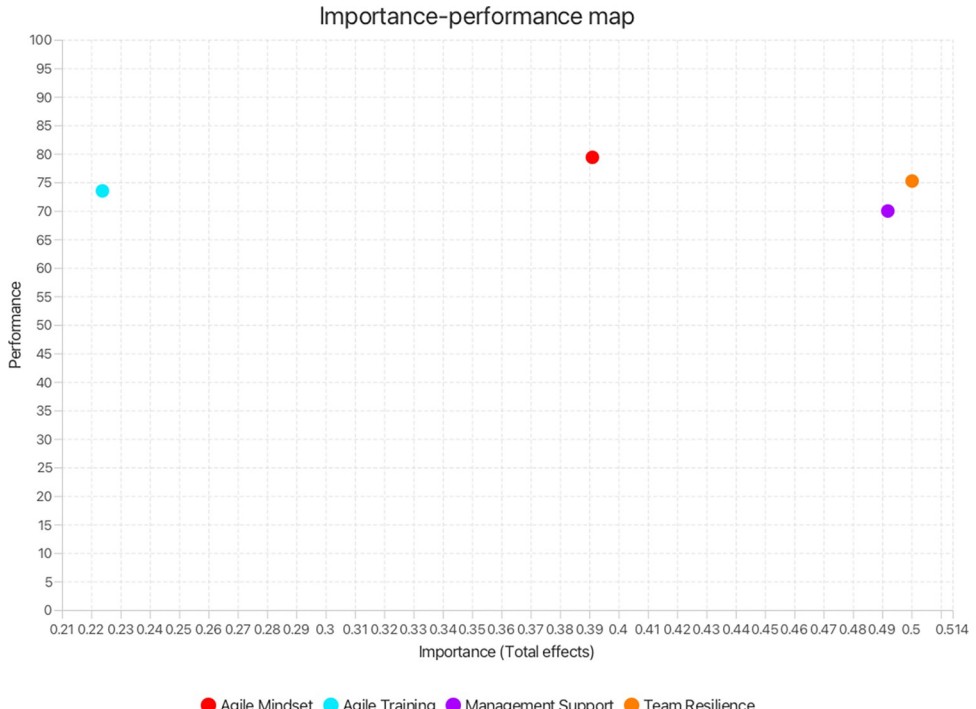

**Fig 4. Importance-performance map at construct level (Source: Authors' calculation).**

found that the PLS-SEM model yielded lower prediction errors on all indicators of SAU, thus outperforming the naïve linear model and establishing high predictive power [38,39].

## 4.3. Results of the importance-performance map analysis

The undertaking of an IPMA necessitates the prior fulfilment of three requirements [40,41]. First, all construct measurements must be undertaken using a metric or quasi-metric scale to enable the rescaling of items from 0–100 [127]. Second, all scales must be calibrated in a single direction where the lowest value represents the least desirable outcome, and the highest value represents the most desirable. Third, it is imperative that outer weights of all indicators of a construct be positive. All three requirements were satisfied, allowing the running of the IPMA. Next, to rescale the latent variable scores from 0 to 100, the theoretical minimum and maximum values of the measurement scales were utilised. Since the items were measured on a 5-point Likert scale ranging from 1 (Strongly Disagree) to 5 (Strongly Agree), the minimum and maximum values were 1 and 5, respectively.

Fig 4 and Table 7 provide a comprehensive representation of the IPMA results at construct level. The analysis revealed that TR in the upper-right quadrant exhibits the highest relative

**Table 7. Importance vs. performance at construct level (Source: Authors' calculation).**

| Construct | Importance (total effects) | Performance (rescaled latent variable scores) | Importance vs. performance |
|---|---|---|---|
| AT | 0.224 | 73.492 | Low importance, low performance |
| AM | 0.391 | 79.373 | Low importance, high performance |
| MS | 0.492 | 69.963 | High importance, low performance |
| TR | 0.500 | 75.214 | High importance, high performance |
| **Average** | **0.402** | **74.511** | |

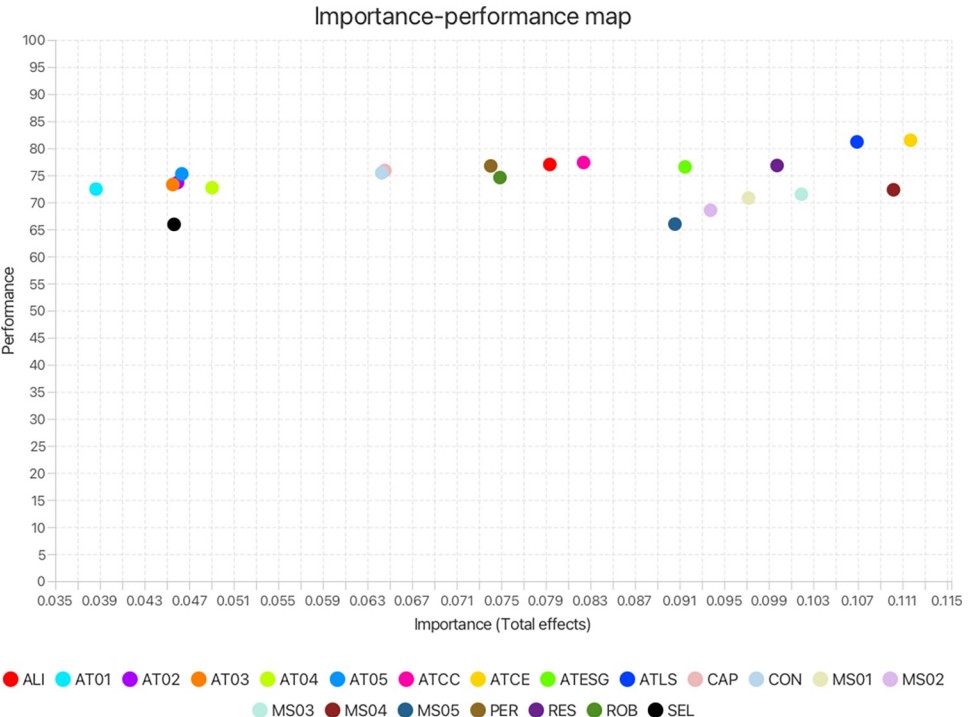

**Fig 5. Importance-performance map at higher-order indicator level (Source: Authors' calculation).**

importance of 0.500, indicating its pivotal role in driving SAU. However, with an above-average performance of 75.214, there is a low potential for further increase. A one-unit increase in TR from 75.214 to 76.214 (as shown in Table 7) can lead to a significant increase of 0.500 points in the performance of SAU. Relative to TR, a one-unit increase of AM in the upper-left quadrant from 79.373 to 80.373 leads to an increase of 0.391 points in SAU's performance, which, while very slightly below average, is still substantial. On the other hand, AT in the lower-left quadrant exhibits trifling importance (0.224), and slightly below average performance in the furtherance of SAU (73.492). Thus, increasing the performance of AT does not lead to as large an increase in SAU. Relative to the other constructs, MS in the lower-right quadrant shows a considerable importance of 0.492, yet its actual performance at 69.963 falls short of the average performance of constructs. This reveals an area of critical concern for practitioners.

The IPMA of MS at the indicator level (Fig 5) corroborates the findings at the construct level. The underperformance of MS is attributable to and reflected in deficiencies across all five of its indicators. This further underpins the vital need for attention to this construct.

## 4.4. Results of the necessary condition analysis

**4.4.1. Analysis of the scatter plots of the exogenous variables.** Consistent with contemporary guidelines [41–44,86], the scatter plots of the unstandardised latent variable scores for each exogenous variable (MS, AT, AM, and TR) in relation to that of SAU were visually examined (Figs 6–9) to gauge the possibility of necessary conditions. These scatter plots were constructed using the theoretical minimum and maximum values of the measurement scales (1–5) as the upper and lower bounds of the axes [43]. Since the study used the same scale for all indicators, the use of unstandardised scores is justified.

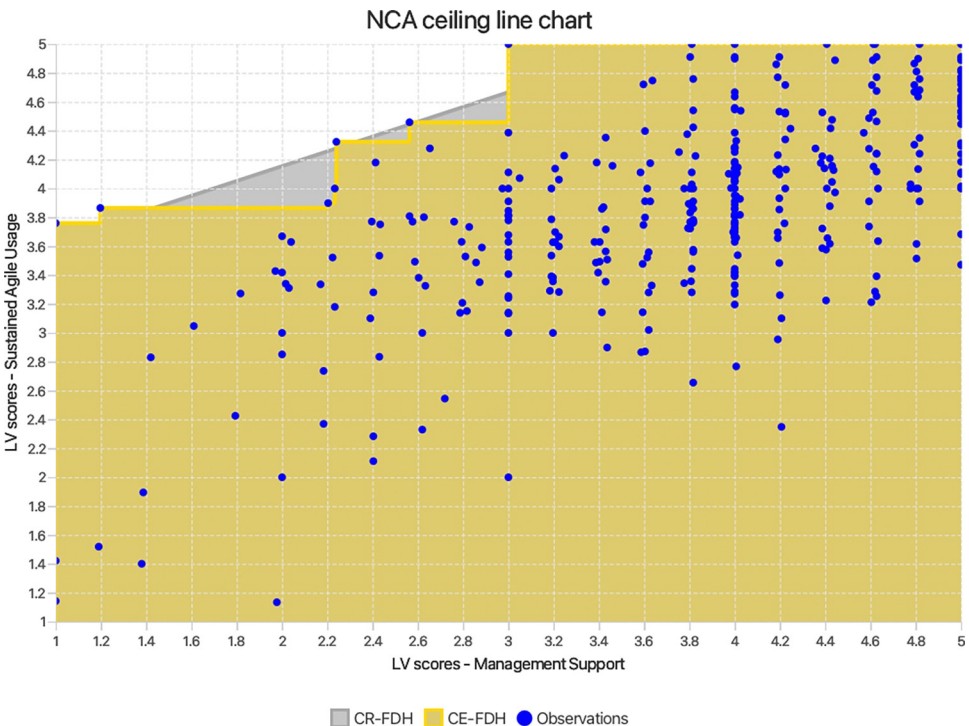

**Fig 6. Scatter plot and ceiling line of MS vs. SAU (Source: Authors' calculation).**

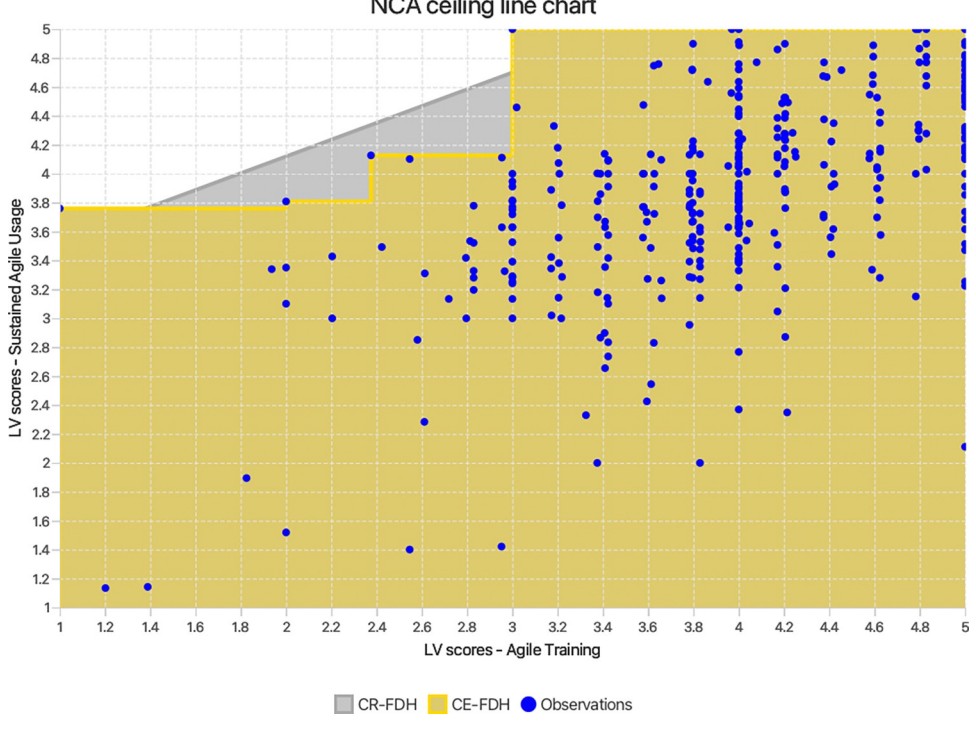

**Fig 7. Scatter plot and ceiling line of AT vs. SAU (Source: Authors' calculation).**

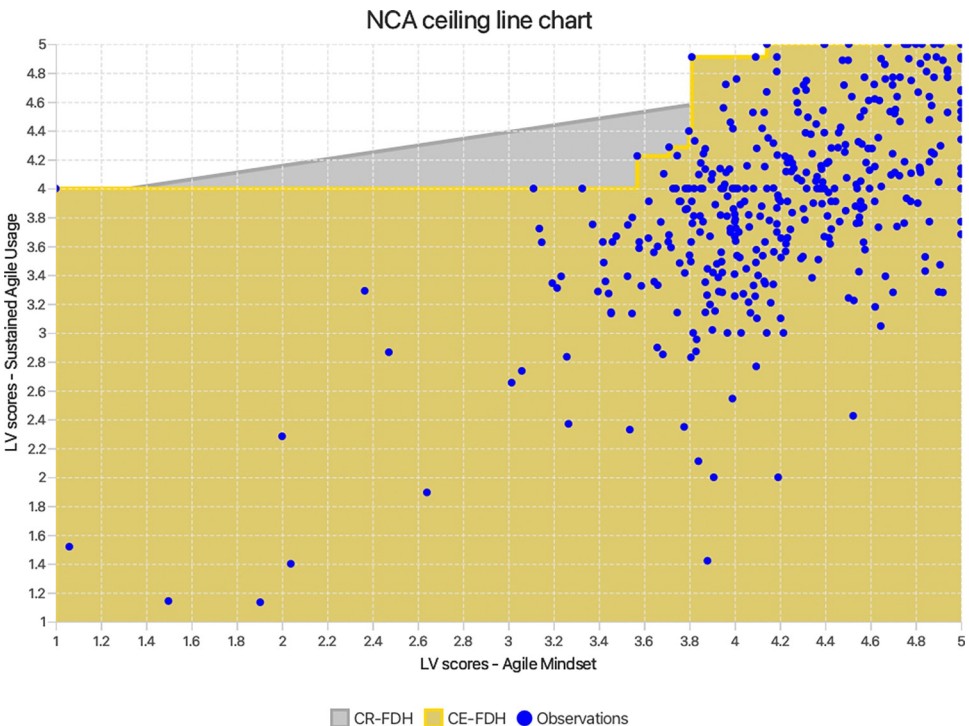

**Fig 8. Scatter plot and ceiling line of AM vs. SAU (Source: Authors' calculation).**

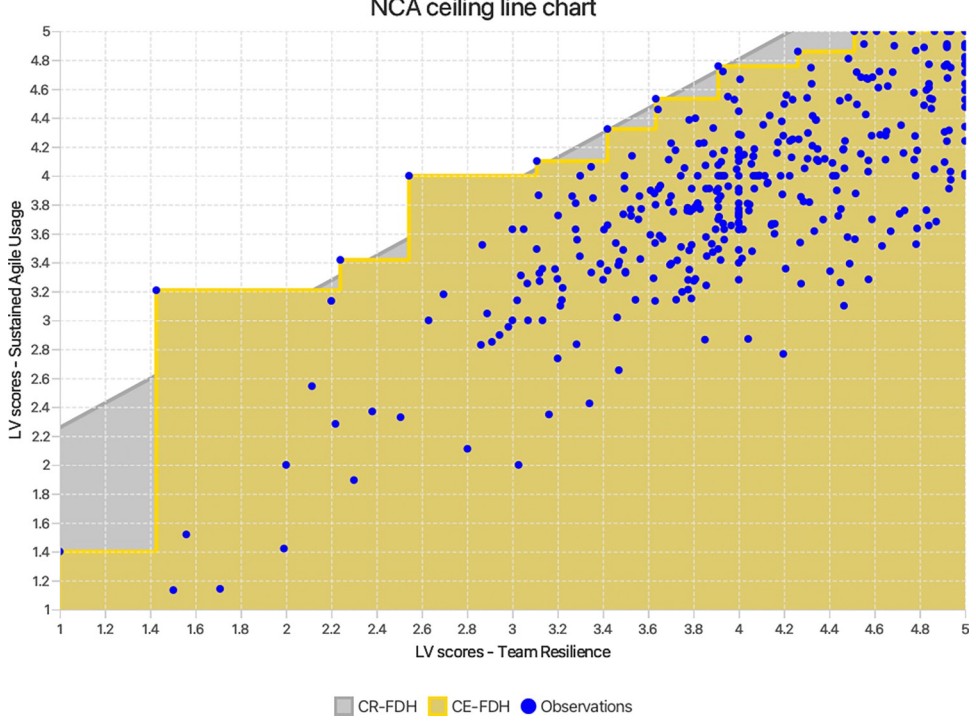

**Fig 9. Scatter plot and ceiling line of TR vs. SAU (Source: Authors' calculation).**

**Table 8. Necessity effect sizes and p-values (Source: Authors' calculation).**

| Hypothesis | Construct | Necessity effect size (CE-FDH line) | Permutation p value | Decision |
|---|---|---|---|---|
| $H_6$ | MS | 0.118 | < 0.001 | Supported |
| $H_7$ | AT | 0.139 | 0.001 | Supported |
| $H_8$ | AM | 0.174 | 0.002 | Supported |
| $H_9$ | TR | 0.294 | < 0.001 | Supported |

Based on the visual check, all four variables showed the possibility of being necessary conditions of SAU, with each having a sizeable area without observations in the upper-left corner. This observation was especially pronounced in TR. Additionally, owing to an irregular observed distribution of data points at the upper border, the ceiling envelopment–free disposable hull line (CE-FDH), that has 100% accuracy by default, was chosen. The CE-FDH line is particularly suitable for ordinal Likert scales with a small number of discrete intervals, such as the ones used in this study [44]. Subsequently, an outlier inspection was carried out using the scatter plots [42,44,127], to reveal several outliers. After a careful examination, these were retained as their existence could not be attributed to a measurement or sampling error [42,104]. Hence, they were determined to be natural variations of the population.

**4.4.2. Effect sizes and significance testing.** Next, the effect size of each exogenous variable was examined, based on the CE-FDH line technique. The results are presented in Table 8. The necessity hypotheses ($H_6$–$H_8$) were assessed at 5% significance, where a hypothesis was rejected if its p-value was greater than 0.05 (insignificant). The authors used the threshold for effect size put forward by Dul [42], where a size of $0 < d < 0.1$, $0.1 <= d < 0.3$, $0.3 <= d < 0.5$ constituted a small effect size, a medium effect size and a large effect size respectively. In accordance with recommendations from past literature [42–44], the minimum effect size required to recognise the practical relevance of a necessary condition was set at 0.1.

The hypotheses $H_6$—$H_9$ were statistically substantiated, with the p-values of each construct below 0.05. All effect sizes surpassed 0.1, thus possessing medium effect size. Thus, it can be concluded that MS, AT, AM and TR all meet the criteria to have a practically significant necessary condition. It is observed that TR has the greatest bottleneck effect of SAU, with an effect size approaching the large category at 0.294. Moreover, it is interesting to note that AT, although of low importance as revealed in the IPMA, was also a necessary condition.

**4.4.3. Bottleneck table analysis.** A bottleneck table is an alternate representation of the ceiling line. The derived results are presented below (Table 9). The level of SAU shows the desired performance of the endogenous construct (latent variable scores rescaled from 0–100). The levels of MS, AT, AM and TR (in bold) represent the performance level of the antecedent construct needed to achieve the desired levels of SAU. The percentiles represent the percentage of cases in the sample that did not meet the bottleneck amount of the exogenous variable to achieve SAU's desired level. 'NN' denotes that the construct does not pose a bottleneck at that particular level of SAU (in other words, it is not a necessary condition at that level).

The results show that until the performance of SAU reaches 10.000, none of the conditions pose bottlenecks on the same. However, at higher levels of SAU, specific thresholds for these constructs become necessary. A minimum level of 15.000 can only be achieved if TR's performance is at 10.662. This shows that TR is a critical determinant at even extremely mild performance levels. Moreover, to achieve a SAU performance level of 70 (at which MS and AT become necessary conditions), thresholds for MS, AT and TR are at 4.921, 25.000 and 38.574 respectively. At a level of 75.000, all variables become necessary conditions, with MS required to be 30.996, AT at 34.374, AM at 64.230 and TR at 52.696.

**Table 9. Bottleneck tables–actual values and percentiles that failed to meet the bottleneck (Source: Authors' calculation).**

| Level of SAU | Level of MS | MS—Percentiles | Level of AT | AT—Percentiles | Level of AM | AM—Percentiles | Level of TR | TR—Percentiles |
|---|---|---|---|---|---|---|---|---|
| 0.000 | NN | 0.000% | NN | 0.000% | NN | 0.000% | NN | 0.000% |
| 5.000 | NN | 0.000% | NN | 0.000% | NN | 0.000% | NN | 0.000% |
| 10.000 | NN | 0.000% | NN | 0.000% | NN | 0.000% | NN | 0.000% |
| 15.000 | NN | 0.000% | NN | 0.000% | NN | 0.000% | 10.662 | 0.256% |
| 20.000 | NN | 0.000% | NN | 0.000% | NN | 0.000% | 10.662 | 0.256% |
| 25.000 | NN | 0.000% | NN | 0.000% | NN | 0.000% | 10.662 | 0.256% |
| 30.000 | NN | 0.000% | NN | 0.000% | NN | 0.000% | 10.662 | 0.256% |
| 35.000 | NN | 0.000% | NN | 0.000% | NN | 0.000% | 10.662 | 0.256% |
| 40.000 | NN | 0.000% | NN | 0.000% | NN | 0.000% | 10.662 | 0.256% |
| 45.000 | NN | 0.000% | NN | 0.000% | NN | 0.000% | 10.662 | 0.256% |
| 50.000 | NN | 0.000% | NN | 0.000% | NN | 0.000% | 10.662 | 0.256% |
| 55.000 | NN | 0.000% | NN | 0.000% | NN | 0.000% | 10.662 | 0.256% |
| 60.000 | NN | 0.000% | NN | 0.000% | NN | 0.000% | 30.992 | 2.558% |
| 65.000 | NN | 0.000% | NN | 0.000% | NN | 0.000% | 38.574 | 3.581% |
| 70.000 | 4.921 | 1.023% | 25.000 | 1.279% | NN | 0.000% | 38.574 | 3.581% |
| 75.000 | 30.996 | 7.161% | 34.374 | 2.813% | 64.230 | 8.951% | 52.696 | 8.696% |
| 80.000 | 30.996 | 7.161% | 50.000 | 7.928% | 64.230 | 8.951% | 60.494 | 15.857% |
| 85.000 | 39.077 | 9.719% | 50.000 | 7.928% | 70.258 | 18.670% | 65.836 | 23.274% |
| 90.000 | 50.000 | 15.345% | 50.000 | 7.928% | 70.258 | 18.670% | 72.734 | 37.340% |
| 95.000 | 50.000 | 15.345% | 50.000 | 7.928% | 70.258 | 18.670% | 81.525 | 63.683% |
| 100.000 | 50.000 | 15.345% | 50.000 | 7.928% | 78.557 | 45.780% | 87.770 | 74.169% |

On a theoretical note, a team wanting to optimise the performance of SAU to 100 would need to ensure that its MS is at 50.000, its AT at 50.000, its members' AM at 78.557 and its TR at a staggering 87.770. However, from a practical perspective, a conservative SAU performance of 80 might be acceptable. This assumed optimal level of 80 demand performances of 30.996 for MS, 50.000 for AT, 64.230 for AM and 60.494 for TR, signifying prominent bottlenecks. Furthermore, it is seen that, at this level, 7.161% of the cases within the sample failed to meet the critical level of MS, while 7.928% of cases did not meet the necessary level of AT. Additionally, a significant portion of 8.951% and 15.857% of cases within the sample failed to meet the threshold of AM and TR respectively. This conveys that a considerable amount of effort should be redirected to ensuring that the antecedent constructs are at an optimal level.

## 5. Discussion

The current study elucidates the complex mechanisms through which MS, AT, AM, and TR foster SAU, thus equipping practitioners and scholars with a better understanding of how to sustain agile practices within ASD teams. Five research questions were sought to be resolved by testing nine hypotheses, to which end it employed a novel methodological approach, combining PLS-SEM, IPMA, and NCA.

With respect to the first question, it was discovered that MS received by an ASD team is a significant predictor of SAU within the team, which resonates considerably with the conclusions of previous researchers [14,24,28,37]. The finding is practically sound due to the fact that effective MS fosters a supportive environment, addresses resource needs, empowers teams, tackles team challenges, builds trust and collaboration, enhances agile behaviour, and provides leadership and guidance, laying the groundwork for successful SAU [14,24,27,28,37].

With respect to the second research question, the analysis reveals that all 03 mediators–AT, AM, and TR–individually mediate the relationship between MS and SAU. The study uncovers that that MS acts primarily through the mediators to impact SAU rather than directly affect SAU, suggesting that these factors work to amplify the impact of the former on the latter. AT equips teams with the necessary skills and knowledge, while the AM fosters a positive and proactive approach to work. TR enables teams to overcome challenges and maintain productivity. These factors are strengthened by the influence of MS, ultimately ensuring that its benefits are fully realised, and SAU is achieved. The findings also highlight the particularly dominant role of TR as an individual mediator in the relationship, explaining over 50% of the mediating effects. This exhibits the considerable need for greater managerial attention towards the construct. Nevertheless, it is observed that all three mediators indisputably contribute to the relationship between MS and SAU, highlighting the importance of considering them in tandem when implementing and sustaining agile practices.

In answering the third research question, the study discovers concrete evidence for a serial mediation effect from MS, through AT, AM and TR, leading to SAU. This is possibly attributable to the mediators' cascading and synergistic influence. Through AT, facilitated by MS, the team members acquire new skills in areas such as collaboration, problem solving, and continuous improvement. This equips them with the tools they need to effectively sustain agile methodologies. At the same time, MS and AT fuels the team's AM, which fosters a culture of openness, innovation, and adaptability. Team members are thus more likely to embrace change, take initiative, and learn from their mistakes. This positive mindset creates a supportive and empowering environment within the team. The combination of AT and the AM contributes to the development of TR. Through acquiring new skills and adopting a positive approach to work while operating in the conducive environment created by MS, team members become better equipped to overcome obstacles and maintain productivity. This resilience is essential for sustaining agile practices in the face of unexpected changes or setbacks. In line with the assertions of Senapathi and Drury-Grogan [14], this finding suggests that managers need to go beyond simply providing resources and actively foster an environment that cultivates agile skills, mindset, and resilience within their team. It is thus evident that enhancing one or more of the antecedents leads to the improvement of subsequent variables, and ultimately the achievement of exponential gains in SAU [28].

Despite the importance of MS within the context of ASD, the undertaken importance-performance map analysis reveals a concerning practical gap between the importance of MS and its actual performance within teams. This evidences that many teams are not receiving the level of MS necessary to fully realise the benefits of agile practices. A deeper investigation revealed that, specifically, the teams felt that they could have benefited from more frequent recognition and appreciation for their work, reduced pressure and criticism, clearer goal-setting guidance, more comprehensive progress tracking, and increased investment in agile training resources.

In response to the final research question, a NCA was undertaken to test previously unconfirmed critical success factors of SAU [14,24,81] through a necessity perspective. It was found that all four antecedents–MS, AT, AM, and TR–were necessary conditions to ensure that SAU performs at the minimum desired level of 80%. It is thus established that inadequate MS, AT, AM and TR prevents the realisation of SAU [14]. In essence, MS, AT, AM and TR are not merely desirable qualities for agile teams but are fundamental for survival and success in today's rapidly changing technological landscape. TR places an exceptionally large bottleneck on the aforementioned, being essential to achieve even 15% performance of SAU. This finding proves plausibly coherent and aligned with prior studies [28], as the absence of an adaptive and resilient capability of an ASD team renders it virtually inept in responding to evolving

needs by changing agile practices accordingly. Without TR, teams become resistant to change, hindering their ability to deliver stakeholder value and ultimately diluting competitive advantage.

Interestingly, placing these findings in context of the previous PLS-SEM and IPMA results, it is observed that there is room for further improvement in all constructs. Although TR was found to be of high importance and performance within ASD teams, 15.857% of the cases did not meet the needed threshold for 80% of SAU, thus warranting managerial efforts towards its optimal maintenance. Given the concerningly low performance of MS as revealed by the IPMA, the discovered necessity of MS underscores the urgent need for teams to take swift action in this area. 7.161% of the cases did not meet the critical MS threshold to ensure 80% performance of SAU. Furthermore, while the previous results may suggest that AT and AM are less critical due to smaller importance or impact, the NCA shows that neglecting them can limit the success of other efforts to improve SAU.

## 5.1. Theoretical implications

This study makes significant contributions to the extant literature on ASD in five ways.

First, this study offers a significant methodological contribution by pioneering the combined use of NCA and IPMA to complement PLS-SEM (cIPMA) within the context of ASD. Thus, it provides evidence for the ability of such techniques to address critical research gaps within ASD research. Through the incorporation of IPMA, the study successfully builds an empirical reference point for how the performance of teams can be assessed. Furthermore, while studies in agile research have identified the necessity of certain factors for SAU, this study is among the first to empirically test these necessity hypotheses. By employing NCA, the study provides concrete evidence for the role of MS, AT, AM, and TR as essential conditions for agile success. Through the fundamental understanding of the relationships thus established by incorporating both sufficiency and necessity logic, the study lays the groundwork for future ASD studies to explore these concepts with even greater nuance. Additionally, the advanced methodological implementation of cIPMA can serve as a reference for other researchers seeking to conduct similar investigations and contribute to the ongoing development of the field.

Second, while existing studies [14,17,24,28,37,37,81,82] have explored the effects of the MS, AT, and AM agile outcomes such as SAU, this study pioneers in providing a novel, comprehensive and integrated framework for understanding their complex interrelationships on the team level. It extends the discourse on SAU as well as its antecedents by incorporating TR, a factor previously unexplored in the context of agile infusion. Using the socio-technical systems theory as a basis, it amalgamates the largely fragmented research on these key constructs to map a more holistic illustration of how the relationships unfold within a team. This cements evidence of their importance in the arena of ASD. On the same note, it pioneers in delivering a convincing explanation of the underlying mechanisms through which MS leads to SAU, providing solid evidence of individual and serial mediation effects within a single framework. It provides deeper insights into the relative importance of their mediating effects, introducing greater nuance of the theoretical relationships. Consequently, the empirical understanding thus derived through the study can contribute massively to the development of novel principles and theories underlying the application of MS, AT, AM and TR in enhancing SAU.

Third, our study responds to a timely call for a re-evaluation of the current understanding of ASD. Given the significant research gap in understanding the post-adoption phase of ASD, this study emerges at a crucial juncture, poised to stimulate a much-needed academic discourse. Providing a comprehensive exploration of distinct factors influencing the post adoption phase, and elucidating the complex interplay between these factors, this research lays the

foundation for future studies in this emerging field. Furthermore, it addresses a critical paucity of literature addressing SAU at the team level, increasing the applicability of post-adoptive agile to the granular level of the ASD team.

Fourth, the study fills a major gap in the literature by providing quantitative evidence to support how the factors of MS, AT, AM and TR interact to foster SAU. While previous research has primarily explored these factors through qualitative methods [14,28,37], this study offers a rigorous quantitative analysis that strengthens our understanding of their roles in maintaining agile practices. The generalisability thus offered is particularly important, given the challenges faced by agile teams in maintaining agile methodologies long term. Furthermore, this quantitative verification of factors affecting SAU will pave the way for future research centred on SAU, stimulating further discourse and development in this underexplored area.

Fifth, while the technical aspects of agile methodologies are often emphasised, the human factors that underpin agile success are equally important. This study contributes to the field by examining the critical role of understudied human factors in sustaining agile usage. By drawing on sociotechnical theory, it operates on a strong theoretical base to explore how the interplay between social factors (MS, AT, AM, TR) impacts the success of the teams' agile practices. Thereby, this study offers a valuable empirical basis for how teams can manipulate these towards developing targeted strategies to enhance agility. Furthermore, it makes a profound contribution to the ongoing discourse on the human dimensions of agile, demonstrating the complexity of the same.

## 5.2. Practical implications and recommendations

The usefulness of this study lies in its meaningful bridging of the gap between theoretical advancement and offering practical guidance for practitioners which are significant and far-reaching. The theoretical framework developed in this study can be applied by practitioners–managers and team members alike–to design and implement targeted interventions to improve SAU. Similarly, the incorporation of formal analyses such as cIPMA offers teams a systematic approach to identify and address performance gaps, enhancing the overall effectiveness of their agile practices. The study demonstrates the practical applicability of cIPMA as a tool for strategic assessment and planning within teams. In other words, the applicability of cIPMA as a powerful tool within teams to identify risks and bottlenecks, and plan resource allocation accordingly is illustrated. This will, in turn guarantee improved and insightful decision making, having a favourable impact on proactive risk mitigation. Thereby, such thorough applications will be monumental in ensuring that the team is receptive to the benefits promised by agile methods, securing competitive advantage.

Therefore, further practical implications and resulting recommendations of the study are hereby expounded separately for both managers and team members. First, it is evident that the top management can play a key role in shaping the team climate such that it is conducive to promote SAU. However, it is also observed that management support is lacking in teams. Thus, this highlights the implication that more effort is required on the part of the top management to facilitate the processes within teams. Managers should therefore ensure a greater acknowledgement of team efforts and contributions, a more supportive and less judgmental environment, more specific and actionable team objectives, regular and detailed monitoring of team performance, and greater allocation of funds and resource people for training and development programs. Furthermore, managers can ensure and reinforce sufficient levels of AT, AM and TR within the team through offering targeted training programs for diverse skills, effective mentorship, greater collaboration opportunities across teams, and rewarding of agile

and resilient behaviour. Moreover, to further enhance a resilient team culture, it is paramount that they facilitate the development of effective problem-solving and decision-making approaches within the team. Managers should also promote a healthy work-life balance by encouraging team members to take breaks and time off when needed while providing resources and support for team members to manage stress and prevent burnout. It is important to note that the failure to ensure the aforementioned can have dire consequences in that it inhibits the genesis of SAU, also dampening the effectiveness of training, the nurturing of the AM within the team, and its ability to adapt and overcome adversity.

Next, the results imply that the commitment of team members to the maintenance of the team's agile culture is of immense value in ensuring that the team's agile practices are adapted as required. As such, ASD team members are urged to seek opportunities to apply the AT received in fulfilling their daily roles within the team, thus reinforcing agile principles and contributing to the team's ongoing success. Furthermore, to cultivate an agile mindset, team members should embrace change, focus on continuous improvement, collaborate effectively, take initiative, think critically and creatively, maintain a positive outlook, seek feedback, and be willing to experiment and learn. By adopting these practices, team members can create a supportive and innovative environment that fosters SAU. To foster a resilient culture the team should ensure that it advances a collective team identity, gives precedence to shared goals over personal agendas, seeks unconventional ways to solve encountered problems incorporating feedback effectively, practices effective prioritisation of tasks and challenges, stays united and motivated in the face of obstacles and consistently leverages the skills and resources at its disposal.

## 6. Conclusion

This study was conducted with the aim of testing the impact of MS on SAU, as well as the individual and serial mediation effects of AT, AM and TR in the aforementioned relationship, through a combined lens of sufficiency and necessity. To this end, the study sought to answer five research questions and validate nine hypotheses. Notably, all five questions were answered, and all nine hypotheses were validated. The key finding of the study was that, on average, the increase of all antecedent variables–MS, AT, AM and TR–leads to an increase of SAU, with AT, AM and TR individually and serially mediating the relationship between MS and SAU. MS and TR are particularly important in ensuring SAU. However, SAU requires a certain minimum level of each antecedent to manifest, calling upon ASD managers and team members to take steps to maintain them as such.

## 7. Limitations and future research

The current study, as any, is constrained by several limitations. First, while it advances literature by examining the roles of AM and TR within ASD, it does not individually examine the roles of their sub-dimensions in fostering SAU. Future research would benefit from a more granular analysis of the sub-dimensions, thus gaining a deeper understanding of their relative importance in fostering SAU. Furthermore, studies examining how these constructs affect diverse aspects of SAU could be additionally beneficial in creating optimal strategies for managerial resource allocation.

Second, the study is of cross-sectional design, impeding definitive conclusions on causality. The authors therefore encourage future researchers to employ longitudinal study designs, allowing for a more comprehensive examination of how the variables evolve over time, and how the mediation effects unfold and strengthen, as well as their temporal sequence and genesis.

Third, it is important to note that the findings may not be fully generalisable to all ASD teams. Further research is needed to ascertain how the cascading influence of MS to AT, AM, TR and SAU, vary across different team sizes, company sizes and other geographical locations. Employing advanced analytical techniques such as multi-group analysis can help identify potential variations in the impact of these factors across differing contexts. Such literary attention would be monumental in providing a more nuanced understanding of the challenges and opportunities facing diverse ASD teams.

Fourth, the study relies on a self-reported questionnaire as a research instrument, introducing the possibility of self-report bias. Future studies could strengthen the validity of these results by employing varied research designs such as observational studies.

## Supporting information

**S1 Data.**
(XLSX)

**S1 Table.**
(DOCX)

## Author Contributions

**Conceptualization:** Uthpala Wijesinghe, Vidara Mapitiyage, Chamoda Wickramage, Krishantha Wisenthige, Chathuni Aluthwala.

**Data curation:** Vidara Mapitiyage, Chathurya Wickramarathne.

**Formal analysis:** Uthpala Wijesinghe.

**Investigation:** Uthpala Wijesinghe.

**Methodology:** Uthpala Wijesinghe, Vidara Mapitiyage, Chathurya Wickramarathne, Chamoda Wickramage, Krishantha Wisenthige.

**Software:** Uthpala Wijesinghe, Chathuni Aluthwala.

**Supervision:** Krishantha Wisenthige, Chathuni Aluthwala.

**Visualization:** Uthpala Wijesinghe, Vidara Mapitiyage.

**Writing – original draft:** Uthpala Wijesinghe, Vidara Mapitiyage, Chathurya Wickramarathne, Chamoda Wickramage.

**Writing – review & editing:** Uthpala Wijesinghe, Chathurya Wickramarathne, Krishantha Wisenthige, Chathuni Aluthwala.

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
