## [Decision Letter · Decision Letter 0]

28 Oct 2024

PONE-D-24-42301Does management support drive sustained agile usage? A serial mediation model and cIPMA perspectivePLOS ONE

Dear Dr. Wisenthige,

Thank you for submitting your manuscript to PLOS ONE. After careful consideration, we feel that it has merit but does not fully meet PLOS ONE’s publication criteria as it currently stands. Therefore, we invite you to submit a revised version of the manuscript that addresses the points raised during the review process. Overall the topic is unique and provides significance contribution to the curren literature. However one of the reviewer has raised some major concerns however the concerns regarding introduction, liteature review and methdology needs to be addressed at priority. The second reviewer have raised minor concerns which should be addressed as well. 

We look forward to receiving your revised manuscript.

Kind regards,

Dr. Farhan Sarwar

Academic Editor

PLOS ONE

Journal Requirements: When submitting your revision, we need you to address these additional requirements. 1. Please ensure that your manuscript meets PLOS ONE's style requirements, including those for file naming. The PLOS ONE style templates can be found at https://journals.plos.org/plosone/s/file?id=wjVg/PLOSOne_formatting_sample_main_body.pdf and https://journals.plos.org/plosone/s/file?id=ba62/PLOSOne_formatting_sample_title_authors_affiliations.pdf 2. In the ethics statement in the Methods, you have specified that verbal consent was obtained. Please provide additional details regarding how this consent was documented and witnessed, and state whether this was approved by the IRB 3. Please include your full ethics statement in the ‘Methods’ section of your manuscript file. In your statement, please include the full name of the IRB or ethics committee who approved or waived your study, as well as whether or not you obtained informed written or verbal consent. If consent was waived for your study, please include this information in your statement as well. 4. Please ensure that you refer to Figure 1 in your text as, if accepted, production will need this reference to link the reader to the figure.

**Additional Editor Comments:**

Dear Authors

The review of the manuscript is complete. You are requested to address the issues raised by the reviewers.

Reviewers' comments:

Reviewer's Responses to Questions

**Comments to the Author**

1. Is the manuscript technically sound, and do the data support the conclusions?

Reviewer #1: Partly

Reviewer #2: Yes

2. Has the statistical analysis been performed appropriately and rigorously? 

Reviewer #1: Yes

Reviewer #2: Yes

3. Have the authors made all data underlying the findings in their manuscript fully available?

Reviewer #1: Yes

Reviewer #2: Yes

4. Is the manuscript presented in an intelligible fashion and written in standard English?

Reviewer #1: Yes

Reviewer #2: Yes

5. Review Comments to the Author

Reviewer #1: Dear Authors

I hope this message finds you well. I would like to express my gratitude for the opportunity to review your insightful manuscript. Engaging with your research has been a rewarding experience, and I appreciate the effort and thoughtfulness that went into your work. I look forward to sharing my feedback.

Reviewer #2: This feedback concerns the manuscript “Does management support drive sustained agile usage? A serial mediation model and cIPMA perspective.” The paper was attractive and well-managed. I appreciate the authors' efforts to conduct empirical research in an area with a considerable gap.

Below are a few suggestions to improve the quality of the work.

1) Avoid using abbreviations in research questions.

2) Minor changes are also recommended in the document.

3) Do not use abbreviations in headings.

4) Do not use abbreviations in hypotheses.

5) H2, H3, and H4 are required to be rewritten or hypothesized again.

6) A stratified multi-stage cluster sampling approach was used and described. However, a description of cluster sampling is missing.

7) An explanation should follow each table. Moreover, Table 3 is described along with Table 2, and the remaining description of Table 2 is given after Table 3.

6. PLOS authors have the option to publish the peer review history of their article (what does this mean?). If published, this will include your full peer review and any attached files.

Reviewer #1: No

Reviewer #2: **Yes: **Dr. Muhammad Adnan Sial

---

## [Author Response · Author response to Decision Letter 0]

18 Nov 2024

We have attached the point to point response to the comments and suggestions as a separate file.

---

## [Editor Report · Decision Letter 1]

1 Dec 2024

PONE-D-24-42301R1Does management support drive sustained agile usage? A serial mediation model and cIPMA perspectivePLOS ONE

Dear Dr. Wisenthige,

Thank you for submitting your manuscript to PLOS ONE. After careful consideration, we invite you to submit a revised version of the manuscript with minor revisions.

The manuscript has immensely improved, and the suggestions from the reviewers are suitably incroporated. There are few minor observations which should be incorporated

Table 6: R2 and F2 are mentioned; the 2 should be in superscript.

Table 7 and Table 9, and their supporting discussion, do not add substantial value to the IPMA (Importance-Performance Map Analysis) section. They do not provide new insights or actionable data beyond what is already discussed. The manuscript is already too detailed. Only IPMA with table 8 is enough, which shows construct-level results. Kindly omit these two tables and reduce the discussion of their result. 

We look forward to receiving your revised manuscript.

Kind regards,

Dr. Farhan Sarwar

Academic Editor

PLOS ONE

Journal Requirements:

Additional Editor Comments:

The manuscript has immensely improved, and the suggestions from the reviewers are suitably incroporated. There are few minor observations which should be incorporated

Table 6: R2 and F2 is mentioned; the 2 should be in superscript.

Table 7 and Table 9, and their supporting discussion do not add substantial value to the IPMA (Importance-Performance Map Analysis) section. If they do not provide new insights or actionable data beyond what is already discussed. The manuscript is already too detailed and it is my suggestion that only IPMA with table 8 is enough, which shows construct-level results.

---

## [Author Response · Author response to Decision Letter 1]

5 Dec 2024

Reviewer Comments 01 

Response to the Comment 01 

We sincerely thank you for bringing this to our attention. 

To address this concern, the reference list was completely reorganised.

First, the following citation (line 1260-1261 of the originally revised manuscript) was removed, and the subsequent references were numbered accordingly. 

5. Rathor S, Batra D, Xia W, Zhang M. What Constitutes Software Development Agility? 2016. 

Next, the reference list was adjusted to reflect the cited order. Thus, the reference below, previously cited under citation number 126 (line 1591-1593 of the originally revised manuscript), was numbered as 93 (line 1516-1518 in the newly revised manuscript). 

93. Wisenthige K. Research Design. In: Saliya CA, editor. Social Research Methodology and Publishing Results: A Guide to Non-Native English Speakers. IGI Global Scientific Publishing; 2023. p. 74–93. doi:10.4018/978-1-6684-6859-3.ch006

Then, each reference was individually updated and made consistent to include complete reference details in the correct formats. For example, issue numbers and page numbers were included where available. The full forms of the journal names were included for clarity. 

The newly reviewed and adjusted reference list can be found on line 1236-1620 in the newly revised manuscript.

Reviewer Comment 02 

Table 6: R2 and F2 is mentioned; the 2 should be in superscript.

Response to the Comment 02

We are grateful for your keen attention to detail. 

We have adjusted the formatting as requested in both the caption and in the table content (line 911 in the newly revised manuscript).

Reviewer Comment 03

Table 7 and Table 9, and their supporting discussion do not add substantial value to the IPMA (Importance-Performance Map Analysis) section. If they do not provide new insights or actionable data beyond what is already discussed. The manuscript is already too detailed and it is my suggestion that only IPMA with table 8 is enough, which shows construct-level results.

Response to the Comment 03

Your insightful comment is greatly appreciated. We acknowledge that Table 7 and Table 9 do not offer substantial additional value. 

In response, the tables 7 and 9 were removed for a more concise yet comprehensive discussion. The subsequent table numbers were adjusted accordingly.

The subheadings ‘Importance-performance map analysis at construct level’ (line 931 of the originally revised manuscript) and ‘Importance-performance map analysis at indicator level’ (line 950 of the originally revised manuscript) were removed. Their contents were merged for better conciseness (line 929-950 in the newly revised manuscript). 

To improve the flow of the newly merged section, the discussion of MS on the construct level was moved from line 937-939 of the originally revised manuscript, to line 939-942 in the newly revised manuscript. This ensures a better continuation of the construct’s discussion on indicator level. 

The fully adjusted section regarding the IPMA on the construct level, on line 929-949 in the newly revised manuscript, reads thus:

“Fig 4 and Table 7 provide a comprehensive representation of the IPMA results at construct level. The analysis revealed that TR in the upper-right quadrant exhibits the highest relative importance of 0.500, indicating its pivotal role in driving SAU. However, with an above-average performance of 75.214, there is a low potential for further increase. A one-unit increase in TR from 75.214 to 76.214 (as shown in Table 7) can lead to a significant increase of 0.500 points in the performance of SAU. Relative to TR, a one-unit increase of AM in the upper-left quadrant from 79.373 to 80.373 leads to an increase of 0.391 points in SAU’s performance, which, while very slightly below average, is still substantial. On the other hand, AT in the lower-left quadrant exhibits trifling importance (0.224), and slightly below average performance in the furtherance of SAU (73.492). Thus, increasing the performance of AT does not lead to as large an increase in SAU. Relative to the other constructs, MS in the lower-right quadrant shows a considerable importance of 0.492, yet its actual performance at 69.963 falls short of the average performance of constructs. This reveals an area of critical concern for practitioners.”

The previous detailed description of the IPMA at the indicator level has been omitted. To still emphasize the importance of scrutinizing the construct MS (as its of low performance yet high importance), the following has been included (line 947-949 in the newly revised manuscript):

“The IPMA of MS at the indicator level (Fig 5) corroborates the findings at the construct level. The underperformance of MS is attributable to and reflected in deficiencies across all five of its indicators. This further underpins the vital need for attention to this construct.”

---

## [Editor Report · Decision Letter 2]

13 Dec 2024

Does management support drive sustained agile usage? A serial mediation model and cIPMA perspective

PONE-D-24-42301R2

Dear Dr. Wisenthige,

We’re pleased to inform you that your manuscript has been judged scientifically suitable for publication and will be formally accepted for publication once it meets all outstanding technical requirements.

Kind regards,

Farhan Sarwar

Academic Editor

PLOS ONE
---

## [Editor Report · Acceptance letter]

19 Dec 2024

PONE-D-24-42301R2 

PLOS ONE

Dear Dr. Wisenthige, 

I'm pleased to inform you that your manuscript has been deemed suitable for publication in PLOS ONE. Congratulations! Your manuscript is now being handed over to our production team.

Kind regards, 

on behalf of

Dr. Farhan Sarwar 

Academic Editor

PLOS ONE